# Guarantees for Alternating Least Squares in Overparameterized Tensor Decomposition

**Dionysis Arvanitakis**
Department of Computer Science
Northwestern University
Evanston, IL 60208
dionarva@u.northwestern.edu

**Vaidehi Srinivas**
Department of Computer Science
Northwestern University
Evanston, IL 60208
vaidehi@u.northwestern.edu

**Aravindan Vijayaraghavan**
Department of Computer Science
Northwestern University
Evanston, IL 60208
aravindv@northwestern.edu

## Abstract

Tensor decomposition is a canonical non-convex optimization problem that is computationally challenging, and yet important due to applications in factor analysis and parameter estimation of latent variable models. In practice, scalable iterative methods, particularly Alternating Least Squares (ALS), remain the workhorse for tensor decomposition despite the lack of global convergence guarantees. A popular approach to tackle challenging non-convex optimization problems is overparameterization— on input an $n \times n \times n$ tensor of rank $r$, the algorithm can output a decomposition of potentially rank $k$ (potentially larger than $r$). On the theoretical side, overparameterization for iterative methods is challenging to reason about and requires new techniques. The work of Wang et al., (NeurIPS 2020) makes progress by showing that a variant of gradient descent globally converges when overparameterized to $k = O(r^{7.5} \log n)$. Our main result shows that overparameterization provably enables global convergence of ALS: on input a third order $n \times n \times n$ tensor with a decomposition of rank $r \ll n$, ALS overparameterized with rank $k = O(r^2)$ achieves global convergence with high probability under random initialization. Moreover our analysis also gives guarantees for the more general low-rank approximation problem. The analysis introduces new techniques for understanding iterative methods in the overparameterized regime based on new matrix anticoncentration arguments.

## 1 Introduction

Iterative heuristics like alternating least squares (ALS), alternate minimization, and gradient descent are the workhorse for many computational tasks in machine learning and high-dimensional data analysis. Their simplicity, scalability and empirical success have led to their widespread use, even for highly non-convex problems. Yet rigorous guarantees have been hard to establish due to the non-convex nature of these problems.

Tensor decomposition is a prime example of a well-studied non-convex problem where there is a disconnect between the practical performance of iterative heuristics and known theoretical guarantees. We are given a third-order tensor $T \in \mathbb{R}^{n \times n \times n}$, and the goal is to decompose the tensor as a sum of

39th Conference on Neural Information Processing Systems (NeurIPS 2025).

a few rank-1 tensors when possible, i.e.,

$$T = \sum_{i=1}^{k} x_i \otimes y_i \otimes z_i,$$

where the $\{x_i \otimes y_i \otimes z_i : i \in [k]\}$ are the rank-1 terms of the decomposition, and the vectors $\{x_i, y_i, z_i : i \in [k]\} \in \mathbb{R}^n$ are called the factors of the decomposition. This is sometimes called the CP decomposition of the tensor, and is an important tool in factor analysis and parameter estimation of many latent variable models in machine learning [see e.g., KB09, Moi18, JGKA19]. Finding the smallest $r$ for which a rank-$r$ decomposition of $T$ exists is NP-hard in the worst case [Hås90, HL13] . Algorithmic guarantees are known under additional genericity or smoothed analysis assumptions for more sophisticated but less scalable algorithms like simultaneous diagonalization and other spectral methods [Har72, LRA93], algebraic methods [DLCC07] and the sum-of-squares hierarchy [BKS15] (see [Vij20] for more detailed comparisons).

The most popular algorithms in practice are iterative methods for the optimization problem given by the least squares objective

$$\min_{\{(x_i,y_i,z_i):i\in[k]\}} \left\| T - \sum_{i=1}^{k} x_i \otimes y_i \otimes z_i \right\|_F^2. \tag{1}$$

In particular, the Alternating Least Squares (ALS) algorithm is an iterative algorithm that alternately updates one set of variables, say $x_1, \ldots, x_k \in \mathbb{R}^n$, while keeping the rest of them ($y_1, \ldots, y_k$ and $z_1, \ldots, z_k$) fixed. Note that each update step is a least squares problem (e.g., in the variables $x_1, \ldots, x_k$) when the remaining variables are fixed (see Section 2 for a more detailed description of the algorithm). The optimization landscape of (1) is highly non-convex, and such iterative algorithms can potentially converge to bad local optima. This has even inspired new variants of ALS like the Orthogonalized ALS algorithm of [SV17], that come with rigorous guarantees under strong assumptions like orthogonality or incoherence of the factors. Yet, the ALS algorithm remains the most popular method for tensor decompositions in practice, despite our poor understanding of when ALS succeeds [KB09, BK25].

Overparameterization has recently emerged as a powerful approach to mitigating non-convexity. Introducing more parameters than those of the ground-truth model often improves optimization dynamics in practice, even in complex settings like training deep neural networks. In our setting, the given tensor $T$ has a rank $r$ decomposition of the form $T = \sum_{i=1}^{r} a_i \otimes b_i \otimes c_i$, and the goal is to find a decomposition of potentially larger rank $k$.

It is challenging to reason about overparameterization with iterative methods for tensor decomposition. Existing approaches based on lazy training and standard mean-field analysis requires an overparameterization of rank that depends polynomially or even exponentially on the ambient dimension. Surprisingly, the work of [WWL+20] makes progress by showing global convergence for a variant of gradient descent with moderate overparameterization of $k = O(r^{7.5} \log n)$ that is nearly independent of the ambient dimension $n$. The main question we address in this paper is:

**Question.** *Does ALS admit a polynomial time global convergence guarantee with moderate overparameterization (a function of $r$ and not $n$)?*

Our main result answers this question in the affirmative. Concretely, our main contributions are the following:

- We prove that the ALS algorithm on a tensor with a mildly conditioned rank-$r$ decomposition[1] when overparameterized with $k = O(r^2)$ and with random initialization of $\{(x_i, y_i, z_i) : i \in [r]\}$, converges with high probability to a global minimum (i.e., objective value 0).

- We also provide rigorous guarantees for ALS under overparameterization for the more general low-rank approximation problem

$$\mathrm{OPT}_r = \min_{\{(x_i,y_i,z_i):i\in[r]\}} \left\| T - \sum_{j=1}^{r} x_i \otimes y_i \otimes z_i \right\|_F^2. \tag{2}$$

---

[1]By midly conditioned we mean that the condition numbers of factor matrices are bounded by $\mathrm{poly}(r)$

We prove that ALS when overparameterized with $k = O(r^2)$ and initialized randomly, finds a solution whose objective value is competitive with the $\text{OPT}_r$ up to a multiplicative factor that is polynomial in only $r$ (and independent of $n$).

For both our results, a moderate overparameterization of $k = O(r^2)$ suffices. We suspect it may be challenging to improve the amount of overparameterization necessary. We remark that even for more computationally-intensive algorithms based on spectral methods, the best polynomial time guarantees require an overparameterization of $k = O(r^2)$ [BCV14, SWZ19]. We leave it as a direction for future work to investigate whether one can prove better upper or lower bounds on the amount of overparameterization necessary to recover a provable guarantee for ALS.

Recent work has developed a few different techniques for analyzing iterative methods with overparameterization. Techniques based on the *lazy training* approach argue that when the model is sufficiently overparameterized, the optimization problem is locally convex and the method will converge to a good solution near the initialization [COB19]. Lazy training analyses incur overparameterization that is polynomial in the ambient dimension $n$, which can be very large compared to the rank $r$.

The work of [WWL$^+$20] makes progress by instead adopting the framework of *mean field analysis*. While previous work that introduced this technique analyzes problems in the case of very large, or even infinite overparameterization [MMN18], [WWL$^+$20] was able to show global convergence for a variant of gradient descent with only moderate overparameterization of $k = O(r^{7.5} \log n)^2$, achieving an exponentially better dependence on $n$ than lazy training analyses for small $r$.

Our work develops an analysis that is significantly different from previous approaches, based on new *matrix anticoncentration* statements. At a high level, we observe that if the iterates $X, Y, Z$ are sufficiently random and in the span of the components of the true tensor, then due to overparameterization they will form a basis for an appropriate space related to the components. If this occurs, then the next iteration of ALS will find a near-exact solution and converge. At initialization, $X, Y, Z$ are independently fully random. However, they are random in the ambient space, and not restricted to the subspace of interest. The first iteration of ALS should in fact update $X, Y, Z$ to be within the correct space. However, the updated $X, Y, Z$ now exhibit significant dependencies on each other. Our analysis shows that despite this, the updated $X, Y, Z$ are still random enough to form the appropriate basis. Thus, the crux of our argument is in showing that this iteration preserves enough randomness from the initialization. Quantitatively, this requires arguing about the *least singular value* of various structured random matrices that arise in the algorithm, along with careful matrix perturbation analyses. Our techniques are applied to a version of ALS that updates all the factor matrices in parallel. Similar guarantees, based on these techniques, for the standard ALS will appear in the arxiv version of the paper.

## 1.1 Related Work

We now describe related work on tensor decompositions and overparameterization, and place our work in the context of these prior works.

Tensor decomposition has a rich history going back to at least [Har70, Har72]. This decomposition into a sum of rank-one tensors is also referred to as CP decomposition or PARAFAC decomposition. See also [KB09] for other decomposition notions for tensors including Tucker decompositions. There are several iterative algorithms that are popular in practice like alternating least squares, alternate minimization, gradient descent and tensor power method [see KB09, AGH$^+$14, JGKA19, for more details]. In particular, the ALS algorithm was first introduced by [Har70, CC70], and has been the workhorse algorithm for tensor decomposition in practice [BK25]. While ALS is popular for its efficiency and steps towards understanding convergence of the iterates have been made [Usc12, WC14], we do not have a good understanding of when it converges to a global optimum solution.

The two results on tensor decompositions that is most relevant to our work are the work of Sharan and Valiant [SV17] who introduced and gave guarantees for an orthogonalized version of ALS, and the work of Wang, Wu, Lee, Ma and Ge [WWL$^+$20] on analyzing gradient descent in the overparameterized regime.

---

[2]For order-$\ell$ tensors the bound on overparameterization is $O(r^{2.5\ell} \log(n))$. Our work only focuses on the order-3 case.

**Comparison to the Orthogonalized ALS algorithm.** The work of Sharan and Valiant [SV17] introduced a variant of the ALS algorithm that orthogonalizes the factors in each step, in addition to the ALS update. As described in [SV17] this allows the ALS algorithm to avoid issues where multiple components of the decomposition capture the same factor of the tensor, when the rank-1 terms have different magnitudes. Their work also proves guarantees under the assumption that the factors $\{a_i : i \in [r]\}$ of the decomposition are orthogonal or incoherent. The decomposition computed by the algorithm is not overparameterized i.e., $k = r$. However, the algorithm is more suited for settings where the target decomposition has near-orthogonal factors. Our analysis is for the ALS algorithm in the overparameterized regime. We do not make incoherence or orthogonality assumptions on the factors; we just need mild conditioning of the factor matrix (condition number that is polynomial in $r$). One can interpret our results as proving that in the overparameterized setting, ALS does not face some fo the earlier issues pointed out in [SV17].

**Overparameterized tensor decompositions using iterative algorithms.** The work of [WWL$^+$20] analyzed a variant of gradient descent in the overparameterized regime of tensor decompositions. Their techniques were able to go beyond lazy-training analyses, and the standard mean-field analysis bounds that require overparameterization of polynomial or even exponential in the ambient dimension $n$. They were surprisingly able to provide guarantees for overparameterized rank $k$ that is almost independent of the ambient dimension. Concretely, their guarantees hold for third-order tensors when the overparameterization is $k = O(r^{7.5} \log n)$; for general order-$\ell$ tensors they need $k = O(r^{2.5\ell} \log n)$. In this work, we instead analyze ALS in the overparameterized regime. We can get guarantees for smaller overparameterized rank $k = O(r^2)$. Moreover, we also approximation guarantees for overparameterized ALS even when the tensor is not exactly of rank $r$. We get guarantees for the more general low-rank approximation problem that incurs a loss that is within a multiplicative factor (depending polynomially only on $r$, and independent of $k$) of the optimum value. To the best of our knowledge, we are unaware of any such guarantees for gradient descent and other iterative algorithms.

**Analysis of other iterative algorithms for tensor decompositions.** There are several other works that try to provide guarantees for iterative methods including alternating minimization, the tensor power method, and other gradient descent based algorithms. The work of [AGH$^+$14] analyzes the tensor power method, and provides guarantees that are specialized to the setting when the factors are orthogonal or near orthogonal [AGH$^+$14, AGJ14, AGJ17]. The works of [JO14, JGKA19] also analyze a variant of the alternating minimization algorithm, and provides convergence guarantees under nearby initialization. These works are not in the overparameterized regime ($k = r$). They find components one at a time but either require stronger assumptions, or provide local convergence guarantees. Finally, it is known that for certain matrix factorization problems and special settings of tensor decomposition (e.g., orthogonal factors), the non-convex optimization landscape is benign i.e., it does not have any local optima that are not globally optimal [Ma21]. However, the general tensor decomposition is highly non-convex with bad local minima as shown in [WWL$^+$20].

**Other tensor decomposition algorithms.** Theoretical guarantees have been established for the simultaneuous diagonalization algorithm [Har72, LRA93, Moi18] and its variants, algebraic methods [DLCC07, JLV23, Koi24, KMW24], sum-of-squares algorithms [BKS15, HSSS16, MSS16]. In the overparameterized setting, spectral methods and algorithms based on subspace embeddings can also find decompositions of rank $k = r^2$ in polynomial time even for the more general low-rank approximation problem with an error that is constant factor competitive with the best rank-$r$ decomposition [SWZ19, BCV14]. The focus of our paper is to prove rigorous guarantees for the ALS algorithm, which is the most popular algorithm in practice.

## 2 Algorithm, Results, and Preliminaries

The Alternating Least Squares (ALS) algorithm for tensor decomposition has many variants. The version that we analyze is given in Algorithm 1 [3]. Given a tensor $T$, the algorithm randomly initializes

---

[3] The version that we analyze is a parallel version of the commonly used ALS method, it however seems like our techniques can be extended to give guarantees for the standard, sequential version of ALS. Our analysis for "sequential" ALS will appear in the arxiv version of the paper.

the three modes $X, Y, Z \in \mathbb{R}^{n \times k}$ of a decomposition, corresponding to a model tensor

$$\widehat{T} = \sum_{i=1}^{k} x_i \otimes y_i \otimes z_i.$$

On each iteration, the algorithm updates each mode individually in parallel to minimize the least squares objective:

$$\min \|T - \widehat{T}\|_F^2.$$

Since $\widehat{T}$ is multilinear in $X, Y, Z$, this is a least squares problem with respect to each mode. The least squares problem could have multiple optima, in fact due to the overparameterization, we expect this to be the case. Typically, ALS is implemented using a linear system solver for each of these subproblems [BK25]; this is also what we will analyze. The updates (Algorithm 1, Lines 8, 12, 16) hence correspond to

$$X^{(t+1)} = \text{flatten}(T, \text{mode } X, \text{modes } Y \otimes Z)(Y^{(t)} \odot Z^{(t)})^{\top^\dagger},$$

$$Y^{(t+1)} = \text{flatten}(T, \text{mode } Y, \text{modes } X \otimes Z)(X^{(t)} \odot Z^{(t)})^{\top^\dagger},$$

$$Z^{(t+1)} = \text{flatten}(T, \text{mode } Z, \text{modes } X \otimes Y)(X^{(t)} \odot Y^{(t)})^{\top^\dagger}.$$

Here we use the shorthand $\text{flatten}(T, \text{mode } A, \text{modes } B \otimes C)$ to mean that the order-3 tensor $T$ is reshaped into a matrix, by taking each $n \times n$ slice in the $B, C$ modes and vectorizing it into an $n^2$-dimensional row of the flattened matrix. There will be $n$ such rows corresponding to mode $A$. (This is explained in more detail in Section 2.1.) Also, $M^\dagger$ refers to the Moore-Penrose pseudoinverse of the matrix $M$.

---

**Algorithm 1** Alternating Least Squares (ALS) for order-3 tensor decomposition

---

**Require:** Tensor $T \in \mathbb{R}^{n \times n \times n}$, rank $r$ of $T$, error tolerance $\varepsilon$
1: $k \leftarrow \Theta(r^2)$     // rank of overparameterized model
2: $X^{(0)} \in \mathbb{R}^{n \times k} \leftarrow \mathcal{N}(0,1)^{n \times k}$, $Y^{(0)} \in \mathbb{R}^{n \times k} \leftarrow \mathcal{N}(0,1)^{n \times k}$, $Z^{(0)} \in \mathbb{R}^{n \times k} \leftarrow \mathcal{N}(0,1)^{n \times k}$
3:     // randomly initialize model
4: $t \leftarrow 0$
5: **while** true **do**
6:     // $X, Y, Z$ updates can be evaluated in parallel
7:     // $X$ update
8:     $\text{err}_X, X^{(t+1)} \leftarrow \min_{X \in \mathbb{R}^{n \times k}}$ and $\arg\min_{X \in \mathbb{R}^{n \times k}}$ of $\left\| T - \sum_{i=1}^{k} X_i \otimes Y_i^{(t)} \otimes Z_i^{(t)} \right\|_F^2$
9:     **if** $\text{err}_X \le \varepsilon$ **then**
10:        **return** $X^{(t+1)}, Y^{(t)}, Z^{(t)}$
11:    // $Y$ update
12:    $\text{err}_Y, Y^{(t+1)} \leftarrow \min_{Y \in \mathbb{R}^{n \times k}}$ and $\arg\min_{Y \in \mathbb{R}^{n \times k}}$ of $\left\| T - \sum_{i=1}^{k} X_i^{(t)} \otimes Y_i \otimes Z_i^{(t)} \right\|_F^2$
13:    **if** $\text{err}_Y \le \varepsilon$ **then**
14:        **return** $X^{(t)}, Y^{(t+1)}, Z^{(t)}$
15:    // $Z$ update
16:    $\text{err}_Z, Z^{(t+1)} \leftarrow \min_{Z \in \mathbb{R}^{n \times k}}$ and $\arg\min_{Z \in \mathbb{R}^{n \times k}}$ of $\left\| T - \sum_{i=1}^{k} X_i^{(t)} \otimes Y_i^{(t)} \otimes Z_i \right\|_F^2$
17:    **if** $\text{err}_Z \le \varepsilon$ **then**
18:        **return** $X^{(t)}, Y^{(t)}, Z^{(t+1)}$
19:    $t \leftarrow t + 1$

---

We give the following guarantee for Algorithm 1. In what follows, we will assume that ALS uses a sub-routine for solving the linear system in polynomial time; concretely, it computes the pseudo-inverse solution up to arbitrary precision $\varepsilon > 0$ in Frobenius norm in time polynomial in $n, \log(1/\varepsilon)$.

**Theorem 2.1** (Guarantee for overparameterized ALS). *For any constant $c_0 > 0$, there exists constants $c = c(c_0) \ge 1$ and $\gamma_0 \in (0, 1)$, such that the following holds. Let $A, B, C \in \mathbb{R}^{n \times r}$ be the factor*

*matrices of the decomposition of a rank-$r$ tensor $T$,*

$$T = \sum_{i=1}^{r} a_i \otimes b_i \otimes c_i,$$

*and suppose the condition numbers $\kappa(A), \kappa(B), \kappa(C) \leq r^{c_0}$. Then, given $T$, an error parameter $\varepsilon$, and a $k \in \mathbb{N}$ satisfying $cr^2 \leq k \leq n^{\gamma_0}$, with probability at least $1 - o(1)$, Algorithm 1 runs in polynomial time and in $O(1)$ steps finds a rank-$k$ decomposition $X, Y, Z \in \mathbb{R}^{n \times k}$ of $T$. That is, $X, Y, Z$ satisfy*

$$\|T - \sum_{i=1}^{k} x_i \otimes y_i \otimes z_i\|_F^2 \leq \varepsilon.$$

The above theorem shows that ALS succeeds from random initialization with overparameterized rank $k = O(r^2)$. For the theorem, we analyze standard Gaussian initializiation, the scale of the random initialization does not matter much. For the above theorem, we assume that the factor matrices $A, B$ and $C$ have condition numbers upper bounded by some large polynomial in $r$. This assumption on the condition numbers is quite mild: for example, it is satisfied w.h.p. for a natural smoothed analysis model.[4] It is weaker than incoherence or orthogonality assumptions, as the vectors in our setting can be quite correlated. Moreover, we believe the assumption to be an artifact of our analysis, and may not be necessary.

Finally, our analysis also implies approximation guarantees for overparameterized ALS with $k = O(r^2)$ under random initialization, in the more general low-rank approximation problem, where $T = \sum_{i=1}^{r} a_i \otimes b_i \otimes c_i + E$, where $\|E\|_F$ is the error.

**Theorem 2.2** (Low-rank tensor approximation using overparameterized ALS). *For any constant $c_0 > 0$, there exists constants $c = c(c_0) \geq 1$ and $\gamma_0 \in (0, 1)$, such that the following holds. Let $A, B, C \in \mathbb{R}^{n \times r}$ be the decomposition of a rank-$r$ tensor $T$,*

$$T = \sum_{i=1}^{r} a_i \otimes b_i \otimes c_i + E,$$

*and suppose the condition numbers $\kappa(A), \kappa(B), \kappa(C) \leq r^{c_0}$. Then, given $T$, $r$, and an error parameter $\varepsilon$, for $cr^2 \leq k \leq n^{\gamma_0}$, with probability at least $1 - o(1)$, Algorithm 1 runs in polynomial time and in $O(1)$ steps finds a rank-$k$ decomposition $X, Y, Z \in \mathbb{R}^{n \times k}$ of $T$. That is, $X, Y, Z$ satisfy*

$$\left\|T - \sum_{i=1}^{k} x_i \otimes y_i \otimes z_i\right\|_F^2 \leq \mathrm{poly}(r)\|E\|_F + \varepsilon.$$

The above theorem gives an ALS guarantee under overparameterization for the optimization problem in (2) in the general setting when $OPT_r > 0$, and generalizes Theorem 2.1 (special case when $OPT_r = 0$). The multiplicative factor loss in the objective compared to $OPT_r$ is polynomial in $r$ and independent of the ambient dimension $n$. To the best of our knowledge, such an approximation guarantee was not known previously for ALS or other iterative algorithms like gradient descent.

## 2.1 Notation and Preliminaries

We now introduce some notation and preliminaries that will be used in the rest of the paper. We refer to a tensor $T$ by its decomposition into *factor matrices*. That is, we associate $T$ with the matrices $X, Y, Z \in \mathbb{R}^{n \times r}$, where $r$ is the rank of $T$ and

$$T = \sum_{i=1}^{r} x_i \otimes y_i \otimes z_i,$$

where $x_i, y_i, z_i$ refer to columns $i$ of $X, Y, Z$ respectively, and $\otimes$ when applied to vectors is the outer product. That is, each component $x_i \otimes y_i \otimes z_i$ is an $n \times n \times n$ tensor. Since each component is an

---

[4]E.g., in the smoothed analysis model, you have an arbitrary matrix which is normalized to have columns of at most unit length, with a random perturbation of length $1/\mathrm{poly}(r)$.

outer product of 3 vectors, this is an *order*-3 tensor. Each direction of the outer product is referred to as a *mode*, and we will sometimes refer to the modes of the tensor by the corresponding factor matrix, i.e., the $X$ mode, $Y$ mode, and $Z$ mode. (These are the analogues of the rows and columns of a matrix/order-2 tensor.) The squared Frobenius norm of a tensor $T$ is the sum of the squares of its entries.

It will also be useful to interact with flattened forms of the tensors we analyze. We use $\odot$ to refer to the Khatri-Rao product of two matrices $Y, Z \in \mathbb{R}^{n \times r}$, which has columns given by

$$(Y \odot Z)_i = \text{vec}(y_i \otimes z_i),$$

where $\text{vec}(\cdot)$ reformats an $n \times n$ matrix as an $n^2$-dimensional vector. This is useful to flatten tensors into matrices. In particular, we have

$$\| \sum_{i=1}^{r} x_i \otimes y_i \otimes z_i \|_F^2 = \| \sum_{i=1}^{r} x_i \otimes \text{vec}(y_i \otimes z_i) \|_F^2 = \left\| X(Y \odot Z)^\top \right\|_F^2.$$

This reshaping into a matrix is exactly what arises in the least squares problem in Algorithm 1 (Lines 8, 12, 16). In Section 2 we describe the flattening operation for a tensor $T$ with factor matrices $X, Y, Z$. The flattening is just a reformatting of the entries of $T$, so computing it does not require explicit access to the factor matrices $X, Y, Z$. However, it is indeed the case that

$$\text{flatten}(T, \text{mode } X, \text{modes } Y \otimes Z) = X(Y \odot Z)^\top,$$

which will be useful for our analysis.

Another useful tensor product on matrices is the Kronecker product which we refer to as $\otimes$.[5] The Kronecker product of two matrices $X \in \mathbb{R}^{n \times r}, Y \in \mathbb{R}^{m \times k}$ is an $nm \times rk$ matrix that satisfies

$$(X \otimes Y)\text{vec}(a \otimes b) = \text{vec}(Xa \otimes Yb), \qquad \forall a \in \mathbb{R}^r, b \in \mathbb{R}^k.$$

(The entries can also be written explicitly as $(X \otimes Y)_{n(i_1-1)+i_2, k(j_1-1)+j_2} = X_{i_1, j_1} Y_{i_2, j_2}$.) Since the columns of a Khatri-Rao product are flattenings of rank-1 matrices, this gives the following identity. For $A \in \mathbb{R}^{n \times r}, B \in \mathbb{R}^{m \times k}$, and $X \in \mathbb{R}^{r \times \ell}, Y \in \mathbb{R}^{k \times q}$, we have

$$(A \otimes B)(X \odot Y) = (AX \odot BY). \tag{3}$$

We will use the Moore-Penrose pseudoinverse $M^\dagger \in \mathbb{R}^{m \times n}$ of a matrix $M \in \mathbb{R}^{n \times m}$. This is defined as a matrix such that

$$\forall x \perp \text{Null}(M), \ Mx = y \Rightarrow M^\dagger y = x, \qquad \forall y \notin \text{Im}(M), \ M^\dagger y = 0.$$

where the nullspace of $M$, $\text{Null}(M)$, is the subspace of vectors mapped to 0 by $M$: $\langle \{x \mid Mx = 0\} \rangle$, and the image of $M$, $\text{Im}(M)$ is the subspace of vectors that can be realized by the linear transformation $M$: $\langle \{y \mid \exists x, Mx = y\} \rangle$. Here and elsewhere we use $\langle \cdot \rangle$ to denote the linear span. There are also a few direct expressions for the pseudoinverse that will be useful to us. For $M \in \mathbb{R}^{n \times r}$, where $r \leq n$, if $M$ is rank $r$ ($M$ is full column rank), we have

$$M^\dagger = (M^\top M)^{-1} M^\top.$$

For $M \in \mathbb{R}^{r \times n}$, where $r \leq n$, if $M$ is rank $r$ ($M$ is full row rank), we have

$$M^\dagger = M^\top (MM^\top)^{-1}.$$

## 3  Analysis of the Algorithm

Fix a tensor $T = \sum_{i=1}^{r} a_i \otimes b_i \otimes c_i$ for some rank-$r$ decomposition $A, B, C \in \mathbb{R}^{n \times r}$. Consider iteration $t$ of ALS on $T$. Without loss of generality, we focus on the update to mode $Z$ (Algorithm 1, Line 8). ALS will converge on this step if $X^{(t)}, Y^{(t)}, Z^{(t+1)}$ fits $T$, i.e., the error between the model tensor and the true tensor is below $\varepsilon$. For the purposes of the overview, we will ignore the $\varepsilon$ error term, and refer to this as perfectly fitting the tensor.

---

[5]This overloads the notation that we use for the outer product on vectors. When applied to vectors we will always mean the outer product. When applied to matrices with dimensions larger than 1 we will always mean the Kronecker product.

The least-squares problem for $Z$ at time $t + 1$ is to reconstruct the *slices* of $T$ as linear combinations of the $X_i^{(t)} \otimes Y_i^{(t)}$. That is, slice $j$ of $T$ is given by

$$T_{:,:,j} = \sum_{i=1}^{r} C_{i,j}(a_i \otimes b_i).$$

The least-squares problem along mode $Z$ is then

$$\min_{Z} \left\| T - \sum_{i=1}^{k} x_i^{(t)} \otimes y_i^{(t)} \otimes z_i \right\|_F^2 = \sum_{j=1}^{n} \min_{Z_{:,j}} \left\| T_{:,:,j} - \sum_{i=1}^{r} Z_{i,j} \left( x_i^{(t)} \otimes y_i^{(t)} \right) \right\|_F^2,$$

which is $n$ independent least-squares problems, one for each slice of $T$ or row of $Z$. Thus, ALS will fit $T$ and converge if, for every slice $j$, $T_{:,:,j}$ is realizable as a linear combination of the $\{x_i^{(t)} \otimes y_i^{(t)} : i \in [k]\}$. Since every slice $j$ of $T$ is a linear combination of $\{a_i \otimes b_i : i \in [r]\}$, a sufficient condition for convergence is that

$$\langle \{a_i \otimes b_i : i \in [r]\} \rangle \subseteq \left\langle \left\{ x_i^{(t)} \otimes y_i^{(t)} : i \in [k] \right\} \right\rangle$$

$$\text{i.e., } \operatorname{colspan}(A \odot B) \subseteq \operatorname{colspan}(X^{(t)} \odot Y^{(t)}). \tag{4}$$

The two lines are reshapings of the same statement. Now suppose for a moment that the columns of $Y^{(t)}$ and $X^{(t)}$ were each drawn independently and randomly from $\operatorname{colspan}(A)$ and $\operatorname{colspan}(B)$ respectively (for example, from standard Gaussians over the $r$-dimensional spaces). Then we would have that since $k \in \Omega(r^2)$, with high probability

$$\operatorname{colspan}(A \otimes B) \subseteq \operatorname{colspan}(X^{(t)} \odot Y^{(t)}), \tag{5}$$

where $A \otimes B$ denotes the Kronecker product of $A$ with itself. Since $\operatorname{colspan}(A \odot B) \subseteq \operatorname{colspan}(A \otimes B)$, because the Khatri-Rao product is a subset of the columns of the Kronecker product, (5) is sufficient to ensure convergence.

Of course, the columns of $X$ and $Y$ are not initialized randomly in the span of $A$ and $B$, instead they are random in the whole $n$-dimensional space. This means that at initialization, each column of $X$ ($Y$) can be thought of as the sum of a random vector in the span of $A$ ($B$), and a component orthogonal to $A$ ($B$). Components orthogonal to the span of $A$ ($B$) only make the Frobenius error of the decomposition higher, since they contribute terms that are orthogonal to $T$. That is, denote $X = \overline{X} + X^{\perp}$, where the columns of $\overline{X}$ are in the column span of $A$, and the columns of $X^{\perp}$ are orthogonal to the column span of $A$. Then

$$\left\| T - \sum_{i=1}^{k} x_i \otimes y_i \otimes z_i \right\|_F^2 = \left\| T - \sum_{i=1}^{k} \overline{x_i} \otimes y_i \otimes z_i \right\|_F^2 + \left\| \sum_{i=1}^{k} x_i^{\perp} \otimes y_i \otimes z_i \right\|_F^2.$$

The first step of ALS (Algorithm 1, Line 8) will set $X$ so that the second term is 0, which since $Y$ and $Z$ are randomly initialized means setting $X^{\perp} = 0$. If the first step of ALS only set $X^{\perp}, Y^{\perp}, Z^{\perp} = 0$ and did not modify $\overline{X}, \overline{Y}, \overline{Z}$, then on the second step (4) would hold, and ALS would converge.

This is however not the case as ALS updates $\overline{X}$ as the minimizer of the least squares objective. Thus $\overline{X}^{(1)} = X^{(1)}$ no longer has independent random columns. Instead it is a function of $X^{(0)}, Y^{(0)}, Z^{(0)}$, and $A, B, C$. Our main technical insight is that, despite $X^{(1)}, Y^{(1)}, Z^{(1)}$ having this complex dependence on each other and $A$, $B$ and $C$, (4) holds with high probability. It is straightforward to show that after the first iteration, each of the factor matrices will be in the span of $A$, $B$ and $C$ respectively, which means that $X^{(1)} \odot Y^{(1)}$ will be in the span of $A \otimes B$. Thus, proving that $X^{(1)} \odot Y^{(1)}$ has rank $r^2$ implies that $\operatorname{colspan}\left( X^{(1)} \odot Y^{(1)} \right) = \operatorname{colspan}\left( A \otimes B \right)$, then condition (4) follows since $\operatorname{colspan}(A \odot B) \subseteq \operatorname{colspan}(A \otimes B)$. This is captured by Theorem 4.1, which is the main technical component of our proof, and the focus of the next section.

## 4  Least Singular Value Bound through Anti-Concentration

For the proofs, we will refer to $X^{(1)}, Y^{(1)}, Z^{(1)}$ as $\hat{X}, \hat{Y}, \hat{Z}$. In this section we give an overview of the proof of our claim that $\hat{X} \odot \hat{Y}$ spans $A \otimes A$ when $X, Y$ and $Z$ are initialized randomly with their

entries being i.i.d. standard Gaussians and, in particular, show that this statement is true in a robust sense. Formal statements of this claim as well as detailed proofs can be found in Appendix C. We give an inverse polynomial in $n$ lower bound on the least singular value of the matrix. Our main technical contribution is proving the following theorem.

**Theorem 4.1.** *Under the assumptions of Theorem 2.1 with probability at least $1 - o(1)$ we have that:*

$$\sigma_{r^2}(\hat{X} \odot \hat{Y}) \geq \frac{1}{n^4 \text{poly}(k)}$$

Assuming that $A$ and $B$ are mildly conditioned we can in fact turn our attention to showing least singular value bounds for the following matrix:

$$\left( (B \odot C)^\top (Y \odot Z)^{\dagger^\top} \right) \odot \left( (A \odot C)^\top (X \odot Z)^{\dagger^\top} \right) \in \mathbb{R}^{r^2 \times k}$$

One main challenge is that the entries of the matrices $(Y \odot Z)^\dagger$ and $(X \odot Z)^\dagger$ are random but highly dependent. While there have been powerful techniques developed recently for proving least singular value bounds of matrices with polynomial entries [BESV24], the random matrix in our setting does not exhibit such structure and is therefore difficult to reason about directly.

We proceed by first arguing about the matrix pseudoinverse by showing in Lemma C.1 that with probability at least $1 - o(1)$,

$$(Y \odot Z)^{\dagger^\top} = \frac{1}{n^2}(Y \odot Z)(I + E_1),$$

where $\|E_1\| \leq O\left( \sqrt{\frac{\log(k)}{n}} + \frac{k \log(k)}{n} \right)$ (and similarly for $(X \odot Z)^{\dagger^\top}$).

The high level intuition is that if the columns of the matrix that we are taking the pseudoinverse of were Gaussian vectors then they would be mostly orthogonal, meaning that the pseudoinverse would be close to the transpose of the matrix. Our analysis shows that the same intuition translates to matrices whose columns are tensor products of Gaussian vectors.

Using that, it suffices, up to $\text{poly}(n)$ factors, to analyze the least singular value of the matrix:

$$L = \left( (B \odot C)^\top (Y \odot Z)(I + E_1) \right) \odot \left( (A \odot C)^\top (X \odot Z)(I + E_2) \right)$$

Furthermore, let:

$$\hat{L} = \left( (B \odot C)^\top (Y \odot Z) \right) \odot \left( (A \odot C)^\top (X \odot Z) \right)$$

Assume for now that we have a guarantee stating that $\sigma_{r^2}(\hat{L}) \geq \frac{1}{\text{poly}(k)}$. As we will discuss later, this is challenging, and much of our technical work is devoted to proving this. We can now use this to prove the existence of a matrix $M$ such that:

$$\hat{L}M = \left( (B \odot C)^\top (Y \odot Z) \right) \otimes \left( (A \odot C)^\top (X \odot Z) \right)$$

Matrix $M$ expresses the columns of the Kronecker product of matrices $(B \odot C)^\top (Y \odot Z)$ and $(A \odot C)^\top (X \odot Z)$ as linear combinations of the columns of their Khatri-Rao product. Such a matrix is guaranteed to exist only because we have assumed that $\hat{L}$ spans $\mathbb{R}^{r^2}$. Using (3) we can express $L$ as:

$$L = \left( (B \odot C)^\top (Y \odot Z)(I + E_1) \right) \odot \left( (A \odot C)^\top (X \odot Z)(I + E_2) \right)$$
$$= \left( (B \odot C)^\top (Y \odot Z) \right) \otimes \left( (A \odot C)^\top (X \odot Z) \right) (I \odot I + E)$$

where $E = I \odot E_2 + E_1 \odot I + E_1 \odot E_2$; further by Lemma C.8, $\|E\| \leq O\left( \sqrt{\frac{\log(k)}{n}} + \frac{\log(k)k}{n} \right)$.

We can now leverage the existence of matrix $M$ to get that:

$$L = \hat{L}(I + ME)$$

where we have crucially used that, by definition of $M$, $M(I \odot I) = I$. In Lemma C.3, we use that $\sigma_{r^2}(\hat{L}) \geq 1/\text{poly}(k)$ to prove that the spectral norm of $M$ is also bounded by a polynomial in $k$. Hence

$$\|ME\| \leq \|M\| \cdot \|E\| \leq \frac{\text{poly}(k)}{n} = o(1),$$

when $n$ is a sufficiently large compared to $k$. We now have that:

$$\sigma_{r^2}(L) = \sigma_{r^2}\left(\hat{L}(I + ME)\right) \geq \sigma_{r^2}(\hat{L}) \cdot \sigma_{\min}(I + ME) \geq \frac{(1 + o(1))}{\text{poly}(k)}$$

thus establishing the main least singular value claim.

In the above proof overview, we assumed a lower bound on the least singular value of $\hat{L}$. The proof of this claim is quite technical and involves a careful net argument along with anti-concentration of low-degree polynomials of independent random variables [CW01]. We first express the columns of the matrix in a convenient way that factors the dependency of having $Z$ on both sides of the Khatri-Rao product. We then argue by applying an $\varepsilon$-net argument and showing that for every fixed vector in $\mathbb{R}^{r^2}$, the probability that the inner products between the fixed vector and all the columns of $\hat{L}$ is negligible is exponentially small in $k$. The formal statement with a detailed proof of it can be found in Lemma C.2.

## 5    Conclusion

Our work proves rigorous polynomial-time global convergence guarantees of the popular ALS method for tensor decomposition with moderate overparameterization of $O(r^2)$. It has been challenging to establish rigorous guarantees for iterative heuristics that are the state-of-the-art in practice. Our analysis is based on new matrix anticoncentration techniques to argue about the iterates, that differs significantly from previous approaches. Our theoretical results on overparameterization are also supported by empirical evaluations in Appendix D. It would be compelling to use these techniques to analyze gradient descent, or prove global convergence guarantees for other non-convex optimization heuristics.

## Acknowledgments

The authors were supported by the NSF-funded Institute for Data, Econometrics, Algorithms and Learning (IDEAL) through the grant NSF ECCS-2216970 and the NSF via grant CCF-2154100. In addition, Vaidehi Srinivas was also partly supported by the Northwestern Presidential fellowship.

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

# A  More Preliminaries

Our analysis involves various operations involving Khatri-Rao products, which are quite challenging to reason about. We also make extensive use of the Kronecker product defined in subsection 2.1, which are more natural products for matrices, and easier to reason about. We have the following simple facts about the Kronecker product:

**Fact A.1.** *Let $A, B \in \mathbb{R}^{n \times k}$, then for the Kronecker product of $A$ and $B$, it holds that:*

*1.*

$$\|A \otimes B\| = \|A\| \cdot \|B\| \tag{6}$$

*2.*

$$\|A \otimes B\|_F = \|A\|_F \cdot \|B\|_F \tag{7}$$

*3.*

$$\sigma_{\min} (A \otimes B) = \sigma_{\min}(A) \cdot \sigma_{min}(B) \tag{8}$$

The proof follows easily by noting that if $U_a \Sigma_a V_a$ and $U_b \Sigma_b V_b$ are the singular value decompositions of $A$ and $B$ respectively then $(U_a \otimes U_b)(\Sigma_a \otimes \Sigma_b)(V_a \otimes V_b)$ is a singular value decomposition of matrix $A \otimes B$.

We now introduce some additional notation that we use in our analysis. For a matrix $A$ we use $\Pi_A$ to denote the projection matrix onto the subspace spanned by the columns of $A$. We use $\lambda_i(A)$ to denote the $i$-th eigenvalue of matrix $A$, we also use $\sigma_i(A)$ to denote the $i$-th singular value of matrix $A$ and $\kappa(A)$ to denote the condition number of $A$. For the factor matrices $A$, $B$ and $C$ of the ground truth we use $\kappa = \max\left(\kappa(A), \kappa(B), \kappa(C)\right)$.

For $X, Y$ and $Z$ being the random initializations of our algorithm we use:

$$\hat{L} = \left((B \odot C)^\top (Y \odot Z)\right) \odot \left((A \odot C)^\top (X \odot Z)\right)$$

and also:

$$\hat{L}_K = \left((B \odot C)^\top (Y \odot Z)\right) \otimes \left((A \odot C)^\top (X \odot Z)\right)$$

notice that $\hat{L}$ is the Khatri-Rao product of matrices $(B \odot C)^\top (Y \odot Z)$ and $(A \odot C)^\top (X \odot Z)$ while $\hat{L}_K$ is their Kronecker product. Furthermore, we use $D_z$ to denote the:

$$D_{z_i} = \text{diag}\left\{\langle c_j, z_i \rangle\right\}_{j=1}^r$$

The following expression is useful in our analysis:

**Lemma A.2.** *For the columns of matrices $(B \odot C)^\top (Y \odot Z)$ and $(A \odot C)^\top (X \odot Z)$ we have that:*

$$(B \odot C)^\top (Y \odot Z)_i = D_{z_i} B^\top y_i \text{ and}$$
$$(A \odot C)^\top (X \odot Z)_i = D_{z_i} A^\top x_i$$

*Proof.* We first observe that the $j$-th entry of vector $(B \odot C)^\top (Y \odot Z)_i$ is given by $\langle b_j, y_i \rangle \langle c_j, z_i \rangle$. The vector whose $j$-th entry is given by $\langle b_j, y_i \rangle$ is $B^\top y_i$. Left multiplying this by $D_{z_i}$ gives the result. $\square$

We now prove some useful claims.

**Claim A.3.** *Given matrices $A \in \mathbb{R}^{n_1 \times m}, B \in \mathbb{R}^{n_2 \times m}$, we have*

$$\min\left\{\|A\|_F \cdot \sigma_m(B), \sigma_m(A) \cdot \|B\|_F\right\} \leq \|A \odot B\|_F \leq \min\left\{\|A\|_F \|B\|, \|A\| \|B\|_F\right\},$$

*where $\|A\| = \sigma_1(A)$ is the spectral norm and $\sigma_r(A)$ is the least singular value of the matrix A.*

*Proof.* We have that

$$\|A \odot B\|_F^2 = \operatorname{tr}\Big((A \odot B)^\top (A \odot B)\Big) = \sum_{i \in [m]} \langle a_i \otimes b_i, a_i \otimes b_i \rangle = \|a_i\|_2^2 \|b_i\|_2^2 \qquad (9)$$

For the upper bound, from (9)

$$\|A \odot B\|_F^2 \leq \max_{i \in [m]} \|b_i\|_2^2 \sum_{i \in [m]} \|a_i\|_2^2 = \|A\|_F^2 \max_{i \in [m]} \|b_i\|_2^2 \leq \|A\|_F^2 \|B\|^2.$$

A similar proof shows that $\|A \odot B\|_F^2 \leq \|B\|_F^2 \cdot \|A\|^2$.

For the lower bound, from (9) we have

$$\|A \odot B\|_F^2 \geq \min_{i \in [m]} \|b_i\|_2^2 \sum_{i \in [m]} \|a_i\|_2^2 = \|A\|_F^2 \min_{i \in [m]} \|b_i\|_2^2 \leq \|A\|_F^2 \cdot \sigma_m(B)^2.$$

$\square$

The following claim shows that for a mildly conditioned decomposition, the Frobenius norm of the ground-truth tensor $T$ can be sandwiched up to $\operatorname{poly}(r)$ factors by the corresponding norms of the factor matrices.

**Claim A.4.** *Let $T = \sum_{i=1}^r a_i \otimes b_i \otimes c_i$. Then*

$$\sigma_r(A)\sigma_r(B)\|C\|_F \leq \|T\|_F \leq \|A\|\|B\|\|C\|_F,$$

*where $\|A\| = \sigma_1(A)$ is the spectral norm and $\sigma_r(A)$ is the least singular value of the matrix $A$. A symmetric statement also holds with the Frobenius norm of $A$ or $B$ instead of $C$ being used.*

*Proof.* First, we note that by considering an appropriate flattening of the tensor,

$$\Big\| \sum_{i=1}^r a_i \otimes b_i \otimes c_i \Big\|_F = \|A(B \odot C)^\top\|_F. \qquad (10)$$

We prove the lower bound first. Note that $A$ has full column rank. Hence

$$\|A(B \odot C)^\top\|_F \geq \sigma_r(A)\|B \odot C\|_F \geq \sigma_r(A)\sigma_r(B)\|C\|_F,$$

by using lower bound in Claim A.3. Similarly for the upper bound, we use the upper bound in Claim A.3 to conclude

$$\|T\|_F = \|A(B \odot C)^\top\|_F \leq \sigma_1(A)\|B \odot C\|_F \geq \|A\|\|B\|\|C\|_F,$$

as required. $\square$

We will use the classic perturbation bound for top-$k$ singular space of a matrix due to Davis and Kahan. The following is a consequence that we use in our robust analysis [see Theorem VII.3.2 Bha97].

**Fact A.5.** *Let $M, \widetilde{M} \in \mathbb{R}^{d \times m}$, and let $\Pi$ denote the projection matrix onto the column space of $M$, and let $\widetilde{\Pi}$ denote the left singular space of $\widetilde{M}$ corresponding to the singular values larger than $\delta$ respectively. Then for a universal constant $c > 0$*

$$\|\widetilde{\Pi}^\perp \Pi\| \leq \frac{c\|M - \widetilde{M}\|}{\sigma_d(M) - \delta}, \qquad (11)$$

*where $\|M\|$ refers to the operator norm of $M$.*

# B  Formal statements of the theorems

We give the formal versions of the main theorems below.

**Theorem B.1** (Guarantee for General decompositions). *For any constant $c_0 > 0$, there exists a constant $c = c(c_0) \geq 1$ such that the following holds. Let $A, B, C \in \mathbb{R}^{n \times r}$ be the decomposition of a rank-$r$ tensor $T$,*

$$T = \sum_{i=1}^{r} a_i \otimes b_i \otimes c_i,$$

*and suppose the condition numbers $\kappa(A), \kappa(B), \kappa(C) \leq r^{c_0} \leq n^{\gamma_0}$ where $\gamma_0 = \gamma_0(c_0)$ is a constant. Then, given $T$, $r$, and an error parameter $\varepsilon$, for $k = C_{op} \cdot r^2$, $C_{op} = C_{op}(c_0)$ is a constnant, Algorithm 1 runs in polynomial time and in $O(1)$ steps finds a rank-$k$ decomposition $X, Y, Z \in \mathbb{R}^{n \times k}$ of $T$, i.e., $\tilde{X}, \tilde{Y}, \tilde{Z}$ satisfy*

$$\left\| T - \sum_{i=1}^{k} \tilde{x}_i \otimes \tilde{y}_i \otimes \tilde{z}_i \right\|_F^2 \leq \varepsilon.$$

As discussed in the proof overview, Theorem B.1 uses the following theorem, generalized for asymmetric decompositions

**Theorem B.2** (Generalization of Theorem 4.1 for asymmetric decompositions). *Under the assumptions of Theorem B.1 with probability at least $1 - o(1)$, we have that:*

$$\sigma_{r^2}\left(\hat{X} \odot \hat{Y}\right) \geq \frac{1}{n^4 \text{poly}(k)}$$

We give a proof of this Theorem in C.1

The following theorem is a robust version of Theorem B.1 and gives a guarantee for low rank approximations

**Theorem B.3** (Guarantee for low-rank approximations). *For any constant $c_0 > 0$, there exists a constant $c = c(c_0) \geq 1$ such that the following holds. Let $A, B, C \in \mathbb{R}^{n \times r}$ be matrices such that for tensor tensor $T$:*

$$T = \sum_{i=1}^{r} a_i \otimes b_i \otimes c_i + Err,$$

*and suppose the condition number $\kappa(A), \kappa(B), \kappa(C) \leq r^{c_0} \leq n^{\gamma_0}$, where $\gamma_0 = \gamma_0(c_0)$ is a constant. Then, given $T$, $r$, and an error parameter $\varepsilon$, for $k = C_{op} r^2$ with $C_{op} = C_{op}(c_0)$ being a constant, Algorithm 1 runs in polynomial time (in $n, k, \log(1/\varepsilon)$ ) and in $O(1)$ steps finds a rank-$k$ decomposition $\tilde{X}, \tilde{Y}, \tilde{Z} \in \mathbb{R}^{n \times k}$ of $T$, i.e., $\tilde{X}, \tilde{Y}, \tilde{Z}$ satisfy*

$$\left\| T - \sum_{i=1}^{k} \tilde{x}_i \otimes \tilde{y}_i \otimes \tilde{z}_i \right\|_F^2 \leq \|Err\|_F^2 \text{poly}(k, r) + \varepsilon.$$

We note that Theorem B.1 is a special case of Theorem B.3. We give a proof of B.3 in subsection C.2; this also implies the correctness of the more specialized Theorem B.1.

# C  Analysis and Proofs

## C.1  Least singular value bound for $\hat{X} \odot \hat{Y}$

In this subsection we give a proof of Theorem B.2 . In particular we show that under the assumptions of Theorem B.1, with probability $1 - o(1)$:

$$\sigma_{r^2}\left(\hat{X} \odot \hat{Y}\right) \geq \frac{1}{n^4 \text{poly}(k)}$$

We use the following lemmas in our analysis.

Lemma C.1 allows us to go from the $(X \odot Z)^{\dagger^\top}$ to matrix $X \odot Z$ times $1/n^2$ multiplied by identity plus an error matrix $E$ whose spectral norm we bound. It is much easier to argue about matrix $\hat{X} \odot \hat{Y}$ when we express $(X \odot Z)^{\dagger^\top}$ and $(Y \odot Z)^{\dagger^\top}$ like that.

**Lemma C.1** (Pseudoinverse transpose simplification). *Let $X, Z \in \mathbb{R}^{n \times k}$ be matrices with i.i.d. standard Gaussian random variables as entries, then there exists an absolute constant $C$ such that, with probability at least $1 - \frac{1}{k}$ :*

$$\left((X \odot Z)^\dagger\right)^\top = \frac{1}{n^2} \cdot (X \odot Z)(I + E)$$

*where $\|E\| \leq 6\sqrt{\frac{C \log(k)}{n}} + \frac{2k}{n} C \log(k)$*

*Proof.* We have that, with probability $1$, the columns of matrix $(X \odot Z)$ are linearly independent, meaning that we can express the pseudoinverse as :

$$(X \odot Z)^\dagger = \left((X \odot Z)^\top (X \odot Z)\right)^{-1} (X \odot Z)^T$$

Taking transpose:

$$\left((X \odot Z)^\dagger\right)^\top = (X \odot Z)\left((X \odot Z)^\top (X \odot Z)\right)^{-1}$$

It therefore suffices to show that:

$$\left(\frac{1}{n^2}(X \odot Z)^\top (X \odot Z)\right)^{-1} = I + E$$

where $E$ has small spectral norm. Let for convenience:

$$W = \frac{1}{n^2} (X \odot Z)^\top (X \odot Z)$$

We show that all the eigenvalues of matrix $W$ are close to $1$ and use that to show that the inverse is close to identity in spectral norm. Specifically, let for convenience:

$$\alpha(n, k) = 3\sqrt{\frac{C \log(k)}{n}} + \frac{k}{n} C \log(k)$$

By Lemma C.4 we have that with probability at least $1 - \frac{1}{k}$, for every eigenvalue of $W$:

$$|\lambda_i(W) - 1| \leq \alpha(n, k)$$

We condition on that event. The eigenvalues of matrix $W^{-1}$ are the reciprocals of the eigenvalues $W$, meaning that assuming that $\alpha(n, k) \leq \frac{1}{2}$, we get that:

$$|\lambda_i(W^{-1}) - 1| = \left|\frac{1}{\lambda_{k-i}(W)} - 1\right|$$
$$= \left|\frac{1}{\lambda_{k-i}(W)}\right| \cdot |\lambda_{k-i}(W) - 1|$$
$$\leq \frac{\alpha(n, k)}{1 - \alpha(n, k)}$$
$$\leq 2\alpha(n, k)$$

Now let $E = W^{-1} - I$ and note that:

$$\{\lambda_i(E)\}_{i=1}^k = \left\{\lambda_i(W^{-1}) - 1\right\}_{i=1}^k$$

Furthermore, $E$ is a symmetric matrix meaning that:

$$\{\sigma_i(E)\}_{i=1}^k = \{|\lambda_i(E)|\}_{i=1}^k$$
$$= \left\{\left|\lambda_i(W^{-1}) - 1\right|\right\}_{i=1}^k$$

which in turn implies that:

$$\|E\| = \sigma_1(E)$$
$$\leq 2\alpha(n, k)$$

the lemma follows. □

Lemma C.2 is the main technical lemma for the proof, it gives us a least singular value bound for matrix $\hat{L}$ and a bound on the Frobenius norm of matrix $\hat{L}_K$. This allows us to bound matrix $M$ by $\text{poly}(k)$. Putting everything together we show that $L = \hat{L}(I + ME)$ where $\hat{L}$ has non-negligible least singular value and $\|ME\| = o(1)$ giving us a non-negligible bound on the least singular value of $L$ which in turn gives us the bound on $\sigma_{r^2}(\hat{X} \odot \hat{Y})$

**Lemma C.2** (Bound on least singular value of $\hat{L}$ and Frob. norm of $\hat{L}_K$)**.** *With probability at least* $1 - \frac{2}{k} - \exp(-4r) - \left(\frac{1}{k}\right)^{r^2}$ *, the following hold:*

1. *For the least singular value of matrix $\hat{L}$, we have that:*

$$\sigma_{r^2}(\hat{L}) \geq \frac{\sigma_r(A)\sigma_r(B)\sigma_r(C)^2}{2C_{CW}^2 k^5 r^2} \tag{12}$$

   *where $C_{CW}$ is an absolute constant coming from the Carbery-Wright inequality, Theorem C.7*

2. *For the matrix:*

$$\hat{L}_K = \left((B \odot C)^\top (Y \odot Z)\right) \otimes \left((A \odot C)^\top (X \odot Z)\right)$$

   *we have that its Frobenius norm is bounded by:*

$$\|\hat{L}_K\|_F \leq 3\tilde{c} \cdot \log(k)\sigma_1(A)\sigma_1(B)\sigma_1(C)^2 r^2 \tag{13}$$

*Proof.* We first focus on item 1. We condition on the event of Lemma C.5 which happens with probability at least $1 - \frac{2}{k}$ and we have that for every $i, j$:

$$\frac{\sigma_r(C)}{k^2 r} \leq |\langle c_j, z_i \rangle| \leq \sqrt{6 \log(k)} \sigma_1(C).$$

We will prove the claim using the variational characterization of singular values, in particular we will show that for very $v \in \mathbb{R}^{r^2}$, we have that:

$$\|\hat{L}v\| \geq \frac{\sigma_r(A)\sigma_r(B)\sigma_r(C)^2}{2C_{CW}^2 k^5 r^2}$$

We consider a fine enough $\varepsilon$-net, $\mathcal{N}$, of $S^{r^2-1}$, where $\varepsilon = \varepsilon(k)$ is an inverse polynomial to be determined later. Note that by Corollary 4.2.13 in [Ver18], we have that $|\mathcal{N}| \leq \left(\frac{3}{\varepsilon}\right)^{r^2}$. Let $u \in \mathcal{N}$, we will bound the probability that:

$$\|\hat{L}^\top u\| \leq \delta \frac{\sigma_r(A)\sigma_r(B)\sigma_r(C)^2}{k^4 r^2}$$

where $\delta$ is a parameter to be specified later. We have that:

$$\mathbb{P}\left(\|\hat{L}^\top u\| \leq \delta \frac{\sigma_1(A)^4}{k^4 r^2}\right) \leq \mathbb{P}\left(\forall i : \left|\langle \hat{L}_i, u \rangle\right| \leq \delta \frac{\sigma_r(A)\sigma_r(B)\sigma_r(C)^2}{k^4 r^2}\right)$$

$$= \prod_i \mathbb{P}\left(\left|\langle \hat{L}_i, u \rangle\right| \leq \delta \frac{\sigma_r(A)\sigma_r(B)\sigma_r(C)^2}{k^4 r^2}\right) \tag{14}$$

where we have used that all the columns of matrix $\hat{L}$ are independent. W now fix an $i$ and bound the probability that $\left|\langle \hat{L}_i, u \rangle\right| \leq \delta \frac{\sigma_r(A)\sigma_r(B)\sigma_r(C)^2}{k^4 r^2}$. We use that, by Lemma A.2, we can express the $i$-th column of matrix $\hat{L}$ as:

$$\hat{L}_i = (D_{z_i} B^\top y_i) \otimes (D_{z_i} A^\top x_i)$$

where $D_{z_i} = \text{diag}\left(\{\langle a_j, z_i \rangle\}\right)$, we can write the inner product with $u$ as:

$$\langle \hat{L}_i, u \rangle = \langle (D_{z_i} B^\top y_i) \otimes (D_{z_i} A^\top x_i), u \rangle$$
$$= \langle \left((D_{z_i} B^\top) \otimes (D_{z_i} A^\top)\right)(y_i \otimes x_i), u \rangle$$
$$= \langle y_i \otimes x_i, (BD_{z_i}) \otimes (AD_{z_i})u \rangle$$

For the least singular value of matrix $BD_{z_i} \otimes AD_{z_i}$ we have that:

$$\sigma_{r^2}\left((BD_{z_i}) \otimes (AD_{z_i})\right) = \sigma_r(BD_{z_i})\sigma_r(AD_{z_i})$$
$$\geq \sigma_r(A)\sigma_r(B)\sigma_r(D_{z_i})^2$$
$$= \sigma_r(A)\sigma_r(B)(\min_j |\langle c_j, z_i \rangle|)^2$$
$$= \frac{\sigma_r(A)\sigma_r(B)\sigma_r(C)^2}{k^4 r^2}$$

let for convenience $v = (AD_{z_i}) \otimes (AD_{z_i})u$, for the norm of $v$ we have that:

$$\|v\| = \|(AD_{z_i}) \otimes (AD_{z_i})u\|$$
$$\geq \frac{\sigma_r(A)\sigma_r(B)\sigma_r(C)^2}{k^4 r^2}.$$

We are interested in bounding the probability that $|\langle y_i \otimes x_i, v \rangle| \leq \delta \frac{\sigma_r(A)\sigma_r(B)\sigma_r(C)^2}{k^4 r^2}$. We will use the Carbery-Wright inequality ([CW01], [Lov10]) to bound the probability. We have that $f(x, y) = \frac{1}{\|v\|}\langle y \otimes x, v \rangle$ is a degree 2 polynomial, satisfying:

$$\text{Var}_{x_i, y_i \sim N(0,1)^n}[f(x_i, y_i)] = 1$$

Applying Theorem C.7 we get that there exists a constant $C_{CW}$:

$$\mathbb{P}\left(|f(x_i, y_i)| \leq \delta\right) \leq C_{CW} \cdot \delta^{1/2}$$

which in turn gives us that:

$$\mathbb{P}\left(|\langle y_i \otimes x_i, v \rangle| \leq \delta \frac{\sigma_r(A)\sigma_r(B)\sigma_r(C)^2}{k^4 r^2}\right) \leq C_{CW} \cdot \delta^{1/2}$$

We can now use equation 14 to get that:

$$\mathbb{P}\left(\|L^\top u\| \leq \delta \frac{\sigma_r(A)\sigma_r(B)\sigma_r(C)^2}{k^4 r^2}\right) \leq \left(C_{CW} \cdot \delta^{1/2}\right)^k$$

We now take a union bound over all elements of the $\varepsilon$-net $\mathcal{N}$, to get that with probability at least $1 - \left(C_{CW} \cdot \delta^{1/2}\right)^k \left(\frac{3}{\varepsilon}\right)^{r^2}$, the bound holds. By Lemma C.6 we have that with probability at least $1 - \exp(-4r)$, for every $i$:

$$\left\|\left((B \odot C)^\top (Y \odot Z)\right)_i\right\| \leq \sqrt{3\tilde{c} \cdot r \log(k)}\sigma_1(A)\sigma_1(C) \text{ and}$$
$$\left\|\left((A \odot C)^\top (X \odot Z)\right)_i\right\| \leq \sqrt{3\tilde{c} \cdot r \log(k)}\sigma_1(A)\sigma_1(C)$$

We use that to bound the spectral norm of $\hat{L}$. We first bound the norm of every column, we have that:

$$\|\hat{L}_i\| = \left\|\left((B \odot C)^\top (Y \odot Z)\right)_i\right\| \cdot \left\|\left((A \odot C)^\top (X \odot Z)\right)_i\right\|$$
$$\leq 3\tilde{c} \cdot r \log(k)\sigma_1(A)\sigma_1(B)\sigma_1(C)^2$$

We now use that the spectral norm is bounded by the Frobenius norm which is in turn bounded by $\sqrt{r}$ times the largest norm of a column to get that:

$$\|\hat{L}\| \leq \|\hat{L}\|_F \leq \sqrt{r} \cdot \max_i \|\hat{L}_i\| \leq 3\tilde{c} \cdot \log(k)\sigma_1(A)\sigma_1(B)\sigma_1(C)^2 r^{3/2}$$

Again applying the union-bound, we get that with probability at least $1 - \exp(-4r) - \left(C_{CW} \cdot \delta^{1/2}\right)^k \left(\frac{3}{\varepsilon}\right)^{r^2}$ both of these events happen. We now take an arbitrary vector $v \in S^{r^2-1}$, we have that there exists a vector $u \in \mathcal{N}$ such that $\|u - v\| \le \varepsilon$:

$$
\begin{aligned}
\|\hat{L}^\top v\| &= \|\hat{L}^\top u + \hat{L}^\top (v - u)\| \\
&\ge \|\hat{L}^\top u\| - \|\hat{L}^\top (v - u)\| \\
&\ge \delta \frac{\sigma_r(A)\sigma_r(B)\sigma_r(C)^2}{k^4 r^2} - \|\hat{L}\|\varepsilon \\
&= \delta \frac{\sigma_r(A)\sigma_r(B)\sigma_r(C)^2}{k^4 r^2} - 3\tilde{c}\log(k)\sigma_1(A)\sigma_1(B)\sigma_1(C)^2 r^{3/2}\varepsilon
\end{aligned}
$$

We take $\delta = \frac{1}{C_{CW}^2 \cdot k}$ and $\varepsilon$ so that we can take $\|\hat{L}v\|$ to be at least half as large as what we get on the $\varepsilon$-net. In particular, we take:

$$
\begin{aligned}
\varepsilon &\le \frac{1}{6C_{CW}^2 \cdot \tilde{c} \cdot k^{2c_0 + 6.75}} \\
&\le \frac{1}{6C_{CW}^2 \cdot \tilde{c} \cdot r^{4c_0} k^5 k^{7/4}} \\
&\le \frac{1}{6C_{CW}^2 \cdot \tilde{c} \cdot \kappa^4 \cdot \log(k) k^5 r^{7/2}} \\
&= \frac{\sigma_r(A)\sigma_r(B)\sigma_r(C)^2}{6C_{CW}^2 \tilde{c}\sigma_1(A)\sigma_1(B)\sigma_1(C)^2 \log(k) k^5 r^{7/2}}
\end{aligned}
$$

we get that:

$$
\begin{aligned}
\|\hat{L}^\top v\| &\ge \frac{\sigma_r(A)\sigma_r(B)\sigma_r(C)^2}{C_{CW}^2 \cdot k^5 r^2} - \frac{3\tilde{c}\log(k)\sigma_1(A)^4 r^{3/2}\sigma_r(A)\sigma_r(B)\sigma_r(C)^2}{6C_{CW}^2 \cdot \tilde{c} \cdot \sigma_1(A)^4 \log(k) k^5 r^{7/2}} = \\
&= \frac{\sigma_r(A)\sigma_r(B)\sigma_r(C)^2}{C_{CW}^2 k^5 r^2} - \frac{\sigma_r(A)\sigma_r(B)\sigma_r(C)^2}{2C_{CW}^2 k^5 r^2} = \\
&= \frac{\sigma_r(A)\sigma_r(B)\sigma_r(C)^2}{2C_{CW}^2 k^5 r^2}
\end{aligned}
$$

The failure probability is upper bounded by:

$$
\frac{2}{k} + \exp(-4r) + (C_{CW}\delta^{1/2})^k \left(\frac{3}{\varepsilon}\right)^{r^2}
$$

We analyze the third term. For convenience, let $C' = 18C_{CW}\tilde{c}$, we have that:

$$
\begin{aligned}
(C_{CW}\delta^{1/2})^k \left(\frac{3}{\varepsilon}\right)^{r^2} &= \left(\frac{1}{k^{1/2}}\right)^k \left(C' k^{2c_0 + 6.75}\right)^{r^2} \\
&= \left(\frac{C' k^{2c_0 + 6.75}}{k^{C_{op}/2}}\right)^{r^2}
\end{aligned}
$$

We can now take $C_{op} = 2(C' + 2c_o + 6.75 + 1)$ to get that:

$$
(C_{CW}\delta^{1/2})^k \left(\frac{3}{\varepsilon}\right)^{r^2} \le \left(\frac{1}{k}\right)^{r^2}.
$$

The first item of the claim follows.

For the second item, recall that we have conditioned on the event that for every $i$:

$$
\begin{aligned}
\left\|\left((B \odot C)^\top (Y \odot Z)\right)_i\right\| &\le \sqrt{3\tilde{c} \cdot r\log(k)}\sigma_1(A)\sigma_1(C) \text{ and} \\
\left\|\left((A \odot C)^\top (X \odot Z)\right)_i\right\| &\le \sqrt{3\tilde{c} \cdot r\log(k)}\sigma_1(A)\sigma_1(C)
\end{aligned}
$$

We have that:

$$\left\|(B \odot C)^\top (Y \odot Z)\right\|_F \leq \sqrt{r} \cdot \max_i \left\|\left((B \odot C)^\top (Y \odot Z)\right)_i\right\|$$

$$\leq \sqrt{3\tilde{c} \cdot \log(k)} \sigma_1(A)\sigma_1(C)r$$

and similarly:

$$\left\|(A \odot C)^\top (X \odot Z)\right\|_F \leq \sqrt{r} \cdot \max_i \left\|\left((A \odot C)^\top (X \odot Z)\right)_i\right\|$$

$$\leq \sqrt{3\tilde{c} \cdot \log(k)} \sigma_1(A)\sigma_1(C)r$$

We can now easily bound the Kronecker product of the two matrices:

$$\left\|\left((B \odot C)^\top (Y \odot Z)^{\dagger^\top}\right) \otimes \left((A \odot C)^\top (X \odot Z)^{\dagger^\top}\right)\right\|_F = \left\|(B \odot C)^\top (Y \odot Z)\right\|_F \cdot \left\|(A \odot C)^\top (X \odot Z)\right\|_F$$

$$\leq 3\tilde{c} \cdot \log(k)\sigma_1(A)\sigma_1(B)\sigma_1(C)^2 r^2$$

$$\square$$

Assuming that the events of Lemma C.2 hold, this lemma bounds the Frobenius norm of matrix $M$:

**Lemma C.3** (Existence and properties of $M$). *Assuming that 12 and 13 of Lemma C.2 hold, then there exists a matrix $M$ such that*

1.
$$\hat{L}M = \hat{L}_K \tag{15}$$

2.
$$M(I \odot I) = I \tag{16}$$

3.
$$\|M\|_F \leq 6C_{CW}^2 \tilde{c}k^5 r^4 \log(k)\kappa^4 + \sqrt{k} \tag{17}$$

*Proof.* We first observe that $\hat{L}$, by Equation 12, spans $\mathbb{R}^{r^2}$ which in particular implies that it spans the columns of matrix $\hat{L}_K$ which lie in this space. This gives us the existence of a matrix $M$ satisfying 15. We now observe that the $i$-th column of matrix $\hat{L}$ is equal to the $((i-1)k+i)$-th column of matrix $\hat{L}_K$. In other words we have that:

$$\hat{L} \cdot e_i = \left(\hat{L}_K\right)_{(i-1)k+i}$$

where $e_i$ is the $i$-th standard basis vector. This implies that we can take:

$$M_{(i-1)k+i} = e_i$$

which in turn gives us that:

$$M(I \odot I) = I$$

and 16 also holds. The rest of the columns of $M$ we select them to be the minimum norm vectors such that $M$ satisfies equation 15. In other words, for $j \neq (i-1)k+i$ for every $i$, we let

$$M_j = \hat{L}^\dagger \left(\hat{L}_K\right)_j$$

We now analyze the Frobenius norm of matrix $M$, defined as above. For $j$ for which there exists an $i$ such that $j = (i-1)k+i$, we have that:

$$\|M_j\| = 1$$

For $j$ such that no such $i$ exists, we have that:

$$\|M_j\| = \left\|\hat{L}^\dagger \left(\hat{L}_K\right)_j\right\|$$

$$\leq \left\|\hat{L}^\dagger\right\| \cdot \left\|\left(\hat{L}_K\right)_j\right\|$$

$$\leq \frac{1}{\sigma_{r^2}\left(\hat{L}\right)} \left\|\left(\hat{L}_K\right)_j\right\|$$

We can now bound the Frobenius norm of matrix $M$:

$$\|M\|_F = \sqrt{\sum_j \|M_j\|^2}$$

$$\leq \sqrt{k + \sum_j \frac{1}{\sigma_{r^2}\left(\hat{L}\right)^2} \left\|\left(\hat{L}_K\right)_j\right\|^2}$$

$$\leq \sqrt{k} + \frac{1}{\sigma_{r^2}\left(\hat{L}\right)} \sqrt{\sum_j \left\|\left(\hat{L}_K\right)_j\right\|^2}$$

$$= \sqrt{k} + \frac{\left\|\hat{L}_K\right\|}{\sigma_{r^2}\left(\hat{L}\right)}$$

where we have used that the square root is a subadditive function. Using the bounds from 12 and 13, we get the result:

$$\|M\|_F \leq \sqrt{k} + 6C_{CW}^2 \tilde{c} \log(k) k^5 r^4 \kappa(A) \kappa(B) \kappa(C)^2$$
$$\leq \sqrt{k} + 6C_{CW}^2 \tilde{c} \log(k) k^5 r^4 \kappa^4$$

$\qquad\qquad\qquad\qquad\qquad\qquad\qquad\qquad\qquad\qquad\qquad\qquad\qquad\qquad\qquad\square$

In Lemma C.4 we argue that the eigenvalues of matrix $(X \odot Z)^\top (X \odot Z)$ are all close to 1 with high probability, we use this to show in Lemma C.1 that we can replace the matrix $(X \odot Y)^{\dagger^\top}$ by $\frac{1}{n^2}(X \odot Y)(I + E)$

**Lemma C.4** (Eigenvalues close to 1). *Let $X, Z \in \mathbb{R}^{n \times k}$ be random matrices with i.i.d standard Gaussian random variables as entries, then with probability at least $1 - \frac{1}{k}$, we have that, for every $i$:*

$$\left| \lambda_i \left( \frac{1}{n^2}(X \odot Z)^\top (X \odot Z) \right) - 1 \right| \leq 3\sqrt{\frac{C \log(k)}{n}} + \frac{k}{n} C \log(k)$$

*where $C$ is an absolute constant*

*Proof.* We will analyze the diagonal and non diagonal entries of the matrix $\frac{1}{n^2}(X \odot Z)^\top (X \odot Z)$ and will show that the diagonal entries are concentrated very close to 1 while the off-diagonal entries are concentrated very close to 0. We will then use Gersgorin disk theorem (Theorem 6.1.1 in [HJ12]) to get the result. For convenience, we let $W = \frac{1}{n^2}(X \odot Z)^\top (X \odot Z)$. For the off-diagonal entries we observe that:

$$W_{i,j} = \frac{1}{n^2} \langle x_i, x_j \rangle \langle z_i, z_j \rangle$$

For the inner product $\langle x_i, x_j \rangle$ we have:

$$\langle x_i, x_j \rangle = \sum_{l=1}^{n} x_i(l) x_j(l)$$

By Lemma 2.2.7 in [Ver18] each summand is a subexponential random variable with subexponential norm:

$$\|x_i(l) x_j(l)\|_{\psi_1} \leq \|x_i(l)\|_{\psi_2} \cdot \|x_j(l)\|_{\psi_2}$$
$$\leq K^2,$$

where $K$ is the sub-Gaussian norm of the standard Gaussian. Using Bernstein's inequality, Theorem 2.8.1 in [Ver18], (the inner product is a sum of sub-exponential random variables) we get that:

$$\mathbb{P}\left( \left| \frac{1}{n} \langle x_i, x_j \rangle \right| \geq \frac{t}{n} \right) \leq 2 \exp\left( -c \min\left( \frac{t^2}{nK^2}, \frac{t}{K} \right) \right)$$

for some absolute constant $c$. Setting $t = \sqrt{C_1 n \log(k)}$, for $C_1$ being a large enough constant we have that with probability at least $1 - \frac{1}{k^3}$:

$$\left| \frac{1}{n} \langle x_i, x_j \rangle \right| \leq \sqrt{\frac{C_1}{n} \log(k)} \tag{18}$$

Similarly, we have that, with probability at least $1 - \frac{1}{k^3}$:

$$\left| \frac{1}{n} \langle z_i, z_j \rangle \right| \leq \sqrt{\frac{C_1}{n} K^2 \log(k^3)} \tag{19}$$

For the diagonal entries of the matrix $M$ we have that:

$$W_{i,i} = \|x_i\|^2 \cdot \|z_i\|^2$$

We write $\|x_i\|^2 = \sum_{l=1}^{n} x_i^2(l)$ and use the fact that there exists an absolute constant $\tilde{C}$ such that $x_i(l)^2 - 1$ is subexponential random variable with subexponential norm:

$$\|x_i^2(l) - 1\|_{\psi_1} \leq \tilde{C} K^2$$

We apply Bernstein's inequality again, to get that:

$$\mathbb{P}\left( \left| \sum_{l=1}^{n} \frac{1}{n} x_i^2(l) - 1 \right| \geq t \right) \leq 2 \exp\left( -c \min\left( \frac{t^2}{\tilde{C} K^2 n}, \frac{t}{K^2} \right) \right)$$

We set $t = \sqrt{C_2 n \log(k)}$ for large enough constant $C_2$, to get that with probability at least $1 - \frac{1}{k^3}$, we have that:

$$\left| \frac{1}{n} \|x_i\|^2 - 1 \right| \leq \sqrt{\frac{C_2 \log(k)}{n}} \tag{20}$$

Similarly, with probability at least $1 - \frac{1}{k^3}$, we have that:

$$\left| \frac{1}{n} \|z_i\|^2 - 1 \right| \leq \sqrt{\frac{C_2 \log(k)}{n}} \tag{21}$$

We get that, by the union bound, with probability at least $1 - \frac{1}{k}$ equations 18, 19, 20 and 21 hold for every $i, j$. Setting $C = \max(C_1, C_2)$, with probability at least $1 - \frac{1}{k}$, for the off-diagonal entries of matrix $(Z \odot X)^\top (Z \odot X)$ we get that:

$$|W_{i,j}| \leq \frac{C \log(k)}{n}$$

And for all diagonal entries, assuming that $\sqrt{\frac{C \log(k)}{n}} \leq 1$, we get that:

$$|W_{i,i} - 1| \leq 3 \sqrt{\frac{C \log(k)}{n}}$$

We now apply Gershgorin disc theorem (Theorem 6.1.1 in [HJ12]) and the the fact that $\frac{1}{n^2} (Z \odot X)^\top (Z \odot X)$ is symmetric and therefore all its eigenvalues are real to get that, for every $i$:

$$\lambda_i(W) \in \bigcup_{j=1}^{n} \left\{ s \in \mathbb{R} : |s - W_{jj}| \leq \sum_{l \neq j} |W_{j,l}| \right\}$$

$$\subseteq \left\{ s \in \mathbb{R} : |s - 1| \leq 3 \sqrt{\frac{C \log(k)}{n}} + \frac{k}{n} C \log(k) \right\}$$

where from the first to the second line we use the triangle inequality. $\qquad \square$

In Lemma C.5 we give bounds upper and lower bounds for the inner products $\langle c_j, z_i \rangle$, needed to bound the least singular value of matrix $\hat{L}$ and the Frobenius norm of matrix $\hat{L}_K$.

**Lemma C.5.** *With probability at least $1 - \frac{2}{k}$ over the randomness in $Z$, we have that for every $i$ and for every $j$:*

$$\frac{\sigma_r(C)}{k^2 r} \leq |\langle c_j, z_i \rangle| \leq \sqrt{6 \log(k)} \sigma_1(C)$$

*Proof.* For fixed $i, j$, the inner product between $c_j$ and $z_i$ has distribution $\langle c_j, z_i \rangle \sim N(0, \|c_j\|^2)$. For the lower bound, we have that:

$$\mathbb{P}\left( |\langle c_j, z_i \rangle| \leq \frac{\sigma_r(C)}{k^2 r} \right) = \mathbb{P}\left( \left| \frac{1}{\|c_j\|} \langle c_j, z_i \rangle \right| \leq \frac{\sigma_r(C)}{k^2 r \cdot \|a_j\|} \right)$$

$$\leq \frac{1}{\sqrt{2\pi}} 2 \frac{\sigma_r(C)}{k^2 r \|c_j\|}$$

$$\leq \frac{\sigma_r(C)}{k^2 r \|c_j\|}$$

where we have used that $\frac{1}{\|c_j\|} \langle c_j, z_i \rangle \sim N(0, 1)$ and that the density of the standard Gaussian is upper bounded by $\frac{1}{\sqrt{2\pi}}$. We now use the union bound to get that:

$$\mathbb{P}\left( \exists i, j : |\langle c_j, z_i \rangle| \leq \frac{\sigma_r(C)}{k^2 r} \right) \leq \sum_{i=1}^{k} \sum_{j=1}^{r} \frac{\sigma_r(C)}{k^2 r \|c_j\|}$$

$$\leq \frac{1}{k}$$

where we have used that for every $j$, $\|c_j\| \geq \sigma_r(C)$. For the upper bound we again fix $i, j$ and use that $\frac{1}{\|c_j\|} \langle c_j, x_i \rangle \sim N(0, 1)$. Using Proposition 2.1.2 in [Ver18], we have that:

$$\mathbb{P}\left( \left| \frac{1}{\|c_j\|} \langle c_j, z_i \rangle \right| \geq t \right) \leq \frac{2}{t} \frac{1}{\sqrt{2\pi}} \exp\left( \frac{-t^2}{2} \right)$$

Letting $t = \sqrt{2 \log(k^3)} \geq 1$, we get that:

$$\mathbb{P}\left( \left| \frac{1}{\|c_j\|} \langle c_j, z_i \rangle \right| \geq \sqrt{2 \log(k^3)} \right) \leq \sqrt{\frac{2}{\pi}} \cdot \frac{1}{k^3} \leq \frac{1}{k^3}$$

By the union bound and the fact that for every $j$, $\|c_j\| \leq \sigma_1(C)$:

$$\mathbb{P}\left( \exists i, j : |\langle c_j, z_i \rangle| \geq \sqrt{2 \log(k^3)} \cdot \sigma_1(C) \right) \leq \sum_{i=1}^{r} \sum_{j=1}^{k} \frac{1}{k^3}$$

$$= \frac{kr}{k^3}$$

$$\leq \frac{1}{k}$$

Using again the union bound we have that with probability at least $1 - \frac{2}{k}$ for every $i, j$:

$$\frac{\sigma_r(C)}{k^2 r} \leq |\langle c_j, z_i \rangle| \leq \sqrt{6 \log(k)} \cdot \sigma_1(C)$$

$\square$

**Lemma C.6** (Bound on columns). *Assume that for every $i, j$, we have that $|\langle c_j, z_i \rangle| \leq \sqrt{6 \log(k)} \sigma_1(C)$, then with probability at least $1 - \exp(-4r)$ we have that for every $i$:*

$$\left\| \left( (B \odot C)^\top (Y \odot Z) \right)_i \right\| \leq \sqrt{\tilde{c} \cdot r \log(k^3)} \sigma_1(B) \sigma_1(C) \quad and$$

$$\left\| \left( (A \odot C)^\top (X \odot Z) \right)_i \right\| \leq \sqrt{\tilde{c} \cdot r \log(k^3)} \sigma_1(A) \sigma_1(C)$$

*where $\tilde{c}$ is a large enough constant.*

*Proof.* Fix an $i$ and recall that by Lemma A.2:

$$\left((A \odot C)^\top (X \odot Z)\right)_i = D_{z_i} A^\top x_i$$

We focus on the norm of $D_{z_i} A^\top x_i$, we have that:

$$D_{z_i} A^\top x_i = D_{z_i} (\Pi_A A)^\top x_i$$
$$= D_{z_i} A^\top \Pi_A x_i$$

where we have used that $\Pi_A A = A$ and that any projection matrix is symmetric. We have that:

$$\|D_{z_i} A^\top x_i\| = \|D_{z_i} A^\top (\Pi_A x_i)\|$$
$$\leq \|D_{z_i} A^\top\| \cdot \|\Pi_A x_i\|$$
$$\leq \|D_{z_1}\| \cdot \|A\| \cdot \|\Pi_A x_i\|$$
$$= \max_j |\langle c_j, z_i \rangle| \cdot \sigma_1(A)\|\Pi_A x_i\|$$
$$\leq \sqrt{2\log(k^3)}\sigma(C)\sigma_1(A)\|\Pi_A x_i\|$$

where from line 2 to line 3 we have used that the operator norm is submultiplicative. From line 3 to line 4 that the operator norm of a diagonal matrix is equal to the largest entry in absolute value and from line 4 to line 5 our assumption on $|\langle c_j, z_i \rangle|$. Similarly, we get that:

$$\|D_{z_i} B^\top y_i\| \leq \sqrt{2\log(k^3)}\sigma_1(B)\sigma_1(C)\|\Pi_B y_i\|$$

We now use that $\|\Pi_A x_i\|$ has the same distribution as that of a norm of an $r$ dimensional random vector with i.i.d. standard Gaussian entries, by Theorem 3.1.1 in [Ver18], we get that:

$$\mathbb{P}\left(\left|\|\Pi_A x_i\| - \sqrt{r}\right| \geq t\right) \leq 2\exp(-ct^2)$$

Applying this bound with $t = (\sqrt{\tilde{c}/2} - 1)\sqrt{r}$ (we will specify $\tilde{c}$ later) as well as the union bound, we get that:

$$\mathbb{P}\left(\exists i : \|\Pi_A x_i\| \geq \sqrt{\frac{\tilde{c}r}{2}} \text{ or } \|\Pi_B y_i\| \geq \sqrt{\frac{\tilde{c}r}{2}}\right) \leq 4k\exp\left(-\left(\sqrt{\tilde{c}/2} - 1\right)^2 r\right)$$

We take $\tilde{c}$ to be a large enough constant so that $4k\exp(-(\sqrt{\tilde{c}/2} - 1)^2 r) \leq \exp(-4r)$. The definition of $k$ in Lemma C.2 depends on $\tilde{c}$, because $C_{op} = 2(18C_{CW} \cdot \tilde{c} + 2c_0 + 7.75)$, we can nevertheless select $\tilde{c}$ large enough so that, for every $r$:

$$2 \cdot (18C_{CW}\tilde{c} + 2c_0 + 8.75) \, r^2 \exp\left(-\left(\sqrt{\tilde{c}/2} - 1\right)^2 r\right) \leq \exp(-4r).$$

We conclude that, with probability at least $1 - \exp(-4r)$, for every $i$:

$$\|D_{z_i} A^\top x_i\| \leq \sqrt{\tilde{c}r\log(k^3)}\sigma_1(A)\sigma_1(C) \text{ and}$$
$$\|D_{z_i} B^\top y_i\| \leq \sqrt{\tilde{c}r\log(k^3)}\sigma_1(B)\sigma_1(C).$$

The claim follows. $\qquad\square$

**Theorem C.7.** *Let $f(x) = f(x_1, x_2, \ldots, x_n)$ be a degree $d$ polynomial such that $Var[f] = 1$ when $x$ is a standard $n$-dimensional Gaussian vector, then for every $t \in \mathbb{R}$ and for every $\varepsilon > 0$, we have that:*

$$\mathbb{P}_{x \sim N(0,1)^n} (|f(x) - t| \leq \varepsilon) \leq O(d) \cdot \varepsilon^{1/d}$$

**Claim C.8.** *Let $A, B \in \mathbb{R}^{n \times k}$, then:*

$$\|A \odot B\| \leq \|A\| \cdot \|B\|$$

*Proof.* We first observe that the columns of $A \odot B$ are a subset of the columns of the matrix $A \otimes B$, meaning that:

$$\|A \odot B\| \leq \|A \otimes B\|$$

we now use that:

$$\|A \otimes B\| = \|A\| \cdot \|B\|$$

$\qquad\square$

**Proof of Theorem B.2** We analyze the least singular value of matrix $\hat{X} \odot \hat{Y}$. Using Equation 3 we can rewrite the matrix $\hat{X} \odot \hat{Y}$ as:

$$\hat{X} \odot \hat{Y} = (A \otimes B)\left(\left((B \odot C)^\top (Y \odot Z)^{\dagger^\top}\right) \odot \left((A \odot C)^\top (X \odot Z)^{\dagger^\top}\right)\right)$$

We now have that:

$$\sigma_{r^2}\left(\hat{X} \odot \hat{Y}\right) \geq \sigma_{r^2}\left(A \otimes B\right)\sigma_{r^2}\left(\left((B \odot C)^\top (Y \odot Z)^{\dagger^\top}\right) \odot \left((A \odot C)^\top (X \odot Z)^{\dagger^\top}\right)\right)$$

$$= \sigma_r\left(A\right)\sigma_r(B)\sigma_{r^2}\left(\left((B \odot C)^\top (Y \odot Z)^{\dagger^\top}\right) \odot \left((A \odot C)^\top (X \odot Z)^{\dagger^\top}\right)\right)$$

We can, without loss of generality, using our assumption on the condition numbers of $A$, $B$ and $C$, assume that for the least singular values of $A$, $B$ and $C$, we have that $\sigma_r(A) \geq \frac{1}{\text{poly}(r)}$, $\sigma_r(B) \geq \frac{1}{\text{poly}(r)}$ and $\sigma_r(C) \geq \frac{1}{\text{poly}(r)}$ (otherwise we can rescale the tensor $T$ so that this holds):

$$\sigma_{r^2}\left(\hat{X} \odot \hat{Y}\right) \geq \frac{1}{\text{poly}(k)} \cdot \sigma_{r^2}\left(\left((B \odot C)^\top (Y \odot Z)^{\dagger^\top}\right) \odot \left((A \odot A)^\top (X \odot Z)^{\dagger^\top}\right)\right)$$

It therefore suffices to analyze the least singular value of matrix:

$$\left((B \odot C)^\top (Y \odot Z)^{\dagger^\top}\right) \odot \left((A \odot C)^\top (X \odot Z)^{\dagger^\top}\right)$$

By Lemma C.1, we have that with probability at least $1 - \frac{2}{k}$:

$$(Y \odot Z)^{\dagger^\top} = \frac{1}{n^2}(Y \odot Z)(I + E_1) \text{ and}$$

$$(X \odot Z)^{\dagger^\top} = \frac{1}{n^2}(X \odot Z)(I + E_2)$$

where $\|E_1\|, \|E_2\| \leq 6\sqrt{\frac{C\log(k)}{n}} + 2C\frac{k}{n}\log(k)$ ($C$ is an absolute constant). We now have that:

$$\left((B \odot C)^\top (Y \odot Z)^{\dagger^\top}\right) \odot \left((A \odot C)^\top (X \odot Z)^{\dagger^\top}\right) = \frac{1}{n^4}L$$

where we have used $L$ to denote:

$$L = \left((B \odot C)^\top (Y \odot Z)(I + E_1)\right) \odot \left((A \odot C)^\top (X \odot Z)(I + E_1)\right)$$

we therefore, have that:

$$\sigma_{r^2}\left(\left((B \odot C)^\top (Y \odot Z)^{\dagger^\top}\right) \odot \left((A \odot C)^\top (X \odot Z)^{\dagger^\top}\right)\right) = \frac{1}{n^4}\sigma_{r^2}(L)$$

Hence, it suffices, in order to prove the claim, to bound the least singular value of matrix $L$ by $\frac{1}{\text{poly}(k)}$. We first analyze the matrix:

$$\hat{L} = \left((B \odot C)^\top (Y \odot Z)\right) \odot \left((A \odot C)^\top (X \odot Z)\right).$$

[6]By Lemma C.2 we have that with probability at least $1 - \frac{2}{k} - \exp(-4r) - \left(\frac{1}{k}\right)^{r^2}$:

$$\sigma_{r^2}\left(\hat{L}\right) \geq \frac{\sigma_r(A)\sigma_r(B)\sigma_r(C)^2}{2C_{CW}^2 k^5 r^2} \tag{22}$$

and that for matrix:

$$\hat{L}_K = \left((B \odot C)^\top (Y \odot Z)\right) \otimes \left((A \odot C)^\top (X \odot Z)\right)$$

Its Frobenius norm is bounded:

$$\left\|\hat{L}_K\right\|_F \leq 3\tilde{c}\log(k)\sigma_1(A)\sigma_1(B)\sigma_1(C)^2 r^2 \tag{23}$$

---

[6]$\hat{L}$ would be equal to $L$ if $E_1 = E_2 = 0$

By Lemma C.3, assuming that equations 22 and 23 hold, there exists a matrix $M$ such that 15, 16 and 17 hold. By the union bound, with probability at least $1 - \frac{4}{k} - \exp(-4r) - \left(\frac{1}{k}\right)^{r^2} = 1 - o(1)$ the events of Lemma C.1 and C.2 both hold. We now have that, using Equation 3:

$$
\begin{aligned}
L &= \left((B \odot C)^\top (Y \odot Z)(I + E_1))\right) \odot \left((A \odot C)^\top (X \odot Z)(I + E_1))\right) \\
&= \left((B \odot C)^\top (Y \odot Z)\right) \otimes \left((A \odot C)^\top (X \odot Z)\right)((I + E_1) \odot (I + E_2)) \\
&= \hat{L}_K (I \odot I + I \odot E_1 + E_2 \odot I) \\
&= \hat{L}_K (I \odot I + E)
\end{aligned}
$$

where we have used $E$ to denote the matrix $I \odot E_2 + E_1 \odot I + E_1 \odot E_2$. By Claim C.8 and assuming that $n$ is large enough compared to $k$, it follows that:

$$
\begin{aligned}
\|E\| &\le \|I \odot E_2\| + \|E_1 \odot I\| + \|E_1 \odot E_2\| \\
&\le \|E_1\| + \|E_2\| + \|E_1\| \cdot \|E_2\| \\
&\le 3\left(6\sqrt{\frac{C\log(k)}{n}} + \frac{2k}{n}C\log(k)\right)
\end{aligned}
$$

We can now use matrix $M$:

$$
\begin{aligned}
L &= \hat{L}_K (I \odot I + E) \\
&= \hat{L}M (I \odot I + E) \\
&= \hat{L}(I + ME)
\end{aligned}
$$

where we have used that $M$ is such that $M(I \odot I) = I$. We will use this expression to analyze the least singular value of matrix $L$. We use the variational characterization of singular values:

$$
\begin{aligned}
\sigma_{r^2}(L) &= \sigma_{r^2}(\hat{L}(I + ME)) \\
&= \sigma_{r^2}\left((I + (ME)^\top)\hat{L}^\top\right) \\
&= \min_{\substack{u \in \mathbb{R}^k, \\ \|u\|=1}} \left\|(I + (ME)^\top)\hat{L}^\top u\right\| \\
&\ge \min_{\substack{u \in \mathbb{R}^k, \\ \|u\|=1}} \sigma_{\min}(I + ME) \cdot \|\hat{L}^\top u\| \\
&\ge \sigma_{\min}(I + ME) \cdot \sigma_{r^2}\left(\hat{L}\right)
\end{aligned}
$$

We have the bound on the least singular value of $\hat{L}$, we only have to analyze the least singular value of matrix $I + ME$. We argue by showing that $ME$ has small spectral norm. We have that:

$$
\|ME\| \le \|M\| \cdot \|E\|
$$

Assuming $n^{\gamma_0} \ge k$ for a small enough constant $\gamma_0$, we get that:

$$
\|M\| \cdot \|E\| = o(1)
$$

Using the variational characterization of the singular values, we have that:

$$
\begin{aligned}
\sigma_{\min}(I + ME) &= \min_{\substack{u \in \mathbb{R}^k, \\ \|u\|=1}} \|(I + ME)u\| \\
&\ge \min_{\substack{u \in \mathbb{R}^k, \\ \|u\|=1}} \|Iu\| - \|MEu\| \\
&\ge 1 - \|ME\| \\
&= 1 - o(1)
\end{aligned}
$$

This concludes the proof.

## C.2 Robust Analysis

Recall that the tensor $T$ has a decomposition of rank $r$ given by factor matrices $A, B, C \in \mathbb{R}^{n \times r}$ that approximates $T$ i.e., $T = \sum_{i=1}^{r} a_i \otimes b_i \otimes c_i + Err$ with $\text{OPT} = \|T - \sum_{i=1}^{r} a_i \otimes b_i \otimes c_i\|_F^2 = \|Err\|_F^2$.

The following lemma relates the objective value to singular values of different matrices related to the updates.

**Lemma C.9.** *[Objective value in the second iteration] Suppose $X^{(1)}, Y^{(1)}$ be the iterates of Algorithm 1 after the updates of the first iteration. Let $\widetilde{\Phi} = X^{(1)} \odot Y^{(1)}$ and $\Phi = \widehat{X} \odot \widehat{Y}$ where $\widehat{X}, \widehat{Y}$ are the updates after the first iteration when there is no error. Then we have that the loss objective value in the second iteration is at most*

$$\left\| T - \sum_{i=1}^{k} x_i^{(1)} \otimes y_i^{(1)} \otimes z_i^{(2)} \right\|_F^2 \leq \frac{O\left( \|\Phi - \widetilde{\Phi}\|^2 \cdot \|T\|_F^2 \right)}{\sigma_{r^2}(\Phi)^2} + 2OPT. \tag{24}$$

*Proof.* We prove this statement using the above lemmas, and using Davis-Kahan theorem for perturbations of top singular spaces.

To bound the objective value in the second iteration as in (24), we use the characterization of least squares value being the squared perpendicular distance of the target vector from the span of the columns i.e., if $\Pi_{\widetilde{\Phi}}^{\perp}$ is the projection matrix onto the subspace orthogonal to the column span of $\widetilde{\Phi} = X^{(1)} \odot Y^{(1)}$, then

$$\left\| T - \sum_{i=1}^{k} x_i^{(1)} \otimes y_i^{(1)} \otimes z_i^{(2)} \right\|_F^2 = \left\| \Pi_{\widetilde{\Phi}}^{\perp} \, \text{flatten}(T, \text{modes } X \otimes Y, \text{mode } Z) \right\|_F^2$$

$$= \left\| \Pi_{\widetilde{\Phi}}^{\perp} \left( (A \odot B)C^\top + E_3 \right) \right\|_F^2,$$

where $E_3 \in \mathbb{R}^{n^2 \times n}$ is the flattening of the tensor $Err$. Let $\Pi_{\Phi}$ be the projection matrix on to the span of the columns of $\widehat{X} \odot \widehat{Y}$, and $\Pi_{\Phi}^{\perp}$ be the projection matrix for the subspace orthogonal to it.

$$\left\| T - \sum_{i=1}^{k} x_i^{(1)} \otimes y_i^{(1)} \otimes z_i^{(2)} \right\|_F^2 = \left\| \Pi_{\widetilde{\Phi}}^{\perp} \left( (A \odot B)C^\top + E_3 \right) \right\|_F^2$$

$$= \left\| \Pi_{\widetilde{\Phi}}^{\perp} (\Pi_{\Phi} + \Pi_{\Phi}^{\perp}) \left( (A \odot B)C^\top + E_3 \right) \right\|_F^2$$

$$\leq 2 \left\| \Pi_{\widetilde{\Phi}}^{\perp} \Pi_{\Phi} \left( (A \odot B)C^\top \right) \right\|_F^2 + 2 \left\| \Pi_{\widetilde{\Phi}}^{\perp} \Pi_{\Phi}^{\perp} \left( (A \odot B)C^\top \right) + \Pi_{\widetilde{\Phi}}^{\perp} E \right\|_F^2$$

$$\leq 2 \left\| \Pi_{\widetilde{\Phi}}^{\perp} \Pi_{\Phi} \left( (A \odot B)C^\top \right) \right\|_F^2 + 2 \|E\|_F^2,$$

where we have used that $\Pi_{\Phi}^{\perp} (A \odot B)C^\top = 0$, since $\Phi = \widehat{X} \odot \widehat{Y}$ contains $A \otimes B$ w.h.p. from the previous non-robust analysis. Furthermore, since the top $r^2$ singular values of $M$ are separated from the least singular value of $\widetilde{\Phi}$ corresponding to $\Pi_{\widetilde{\Phi}}^{\perp}$ (i.e., 0, since this corresponds to the nullspace of $\widetilde{\Phi}$), we have by the Davis-Kahan theorem (see Fact A.5)

$$\|\Pi_{\widetilde{\Phi}}^{\perp} \Pi_{\Phi}\| \leq \frac{O(\|\Phi - \widetilde{\Phi}\|)}{\sigma_{r^2}(\Phi)}.$$

$$\left\| T - \sum_{i=1}^{k} x_i^{(1)} \otimes y_i^{(1)} \otimes z_i^{(2)} \right\|_F^2 \leq 2 \|\Pi_{\widetilde{\Phi}}^{\perp} \Pi_{\Phi}\| \left\| \left( (A \odot B)C^\top \right) \right\|_F^2 + 2\|E\|_F^2$$

$$\leq \frac{c\|\Phi - \widetilde{\Phi}\|^2 \cdot \|T\|_F^2}{\sigma_{r^2}(\Phi)^2} + 2OPT,$$

for some constant $c > 0$. $\qquad \square$

The following claims bound the different terms in (24) Lemma C.11.

**Lemma C.10.** *[Projections onto the column space of $Y \odot Z$] Let $Q \in \mathbb{R}^{n^2 \times r}$ be an arbitrary matrix and $Y, Z \in \mathbb{R}^{n \times k}$ be random matrices with i.i.d $N(0,1)$ entries. There exists a universal constant $c > 0$ such that with probability at least $1 - o(1)$*

$$\left\| Q^\top \left( (Y \odot Z)^\dagger \right)^\top \right\|_F \leq \left( \frac{c\sqrt{k} \log(kr)}{n^2} \right) \cdot \|Q\|_F. \tag{25}$$

*Proof.* From Lemma C.1 we have that

$$((Y \odot Z)^\dagger)^\top = \frac{1}{n^2} (Y \odot Z)(I + E), \text{ where } \|E\| = o_{k,n}(1) \leq 1$$

for our choice of parameters. Hence, it will suffice to upper bound $\|Q^\top (Y \odot Z)\|_F$.

Each of the $k$ columns of $Y \odot Z$ is an i.i.d. random vector distributed identically to $y \otimes z$ where $y, z \sim N(0, I_{n \times n})$. Consider a fixed $j \in [r], i \in [k]$. For any $t \geq 1$, from concentration of quadratic multivariate polynomials due to Hanson-Wright inequality [see e.g., Ver18], we have that the $(j,i)$th entry of $Q^\top(Y \odot Z)$ can be written as

$$\mathbb{P}_{Y,Z} \left[ |\langle Q_j, (Y \odot Z)_i \rangle| > t\|Q_j\|_F \right] = \mathbb{P}_{y,z \sim N(0,I_n)} \left[ |y^\top Q_j z| > t\|Q_j\|_F \right]$$

$$\leq 2 \exp \left( -\frac{t^2 \|Q_j\|_F^2}{2\|Q_j\|_F^2 + t\|Q_j\|} \right) \leq 2 \exp \left( -\frac{t}{3} \right) \leq \frac{1}{3(kr)^2},$$

for $t = O(\log(kr))$. The lemma follows after a union bound to get an upper bound on the magnitude of each of the $kr$ entries of the matrix. $\qquad\square$

This in turns leads to the following claim.

**Lemma C.11.** *[Perturbation of the spectrum with noise] Let $X^{(1)}, Y^{(1)}$ be the iterates of Algorithm 1 after the updates of the first iteration. Let $\widetilde{\Phi} = X^{(1)} \odot Y^{(1)}$ and $\widehat{X}, \widehat{Y}$ denote the updates after the first iteration when there is no error. Then with high probability $1 - o(1)$, we have for $\Phi = \widehat{X} \odot \widehat{Y}$ that*

$$\left\| \widetilde{\Phi} - \Phi \right\| = \left\| X^{(1)} \odot Y^{(1)} - \widehat{X} \odot \widehat{Y} \right\| \leq \frac{\sqrt{OPT} \cdot O(k \log^2(kr))}{n^4} \left( 2\|A\|\|B\|\|C\|_F + \sqrt{OPT} \right). \tag{26}$$

*A similar claim holds for $Y^{(1)} \odot Z^{(1)}$ and $Z^{(1)} \odot X^{(1)}$ as well.*

In the above expression $\sqrt{OPT} \ll \|A\|\|B\|\|C\|_F$.

*Proof.* Recall that $T = \sum_{i=1}^r a_i \otimes b_i \otimes c_i + Err$, where $\|Err\|_F = \sqrt{OPT}$. Moreover, for the flattening $E_1, E_2 \in \mathbb{R}^{n \times n^2}$ along the first and second modes of $Err$, we have with probability $1 - o(1)$ from Lemma C.10 that that

$$X^{(1)} = \left( A(B \odot C)^\top + E_1 \right) \left( (Y \odot Z)^\dagger \right)^\top$$
$$= A(B \odot C)^\top + E_1 \left( (Y \odot Z)^\dagger \right)^\top$$
$$= \widehat{X} + E_1((Y \odot Z)^\dagger)^\top$$
$$= \widehat{X} + E_X,$$

$$\text{where } \|E_X\| \leq \|E_X\|_F \leq \sqrt{OPT} \cdot \frac{\sqrt{k} \log(kr)}{n^2}.$$

$$\text{And similarly, } Y^{(1)} = \widehat{Y} + E_Y, \text{ where } \|E_Y\|_F \leq \sqrt{OPT} \cdot \frac{\sqrt{k} \log(kr)}{n^2}$$

Furthermore,

$$\begin{aligned}
\|\widehat{X}\|_F &= \left\| A(B \odot C)^\top ((Y \odot Z)^\dagger)^\top \right\|_F \\
&\leq \|A\| \cdot \| (B \odot C)^\top \left( (Y \odot Z)^\dagger \right)^\top \|_F \\
&\leq \left( \frac{c\sqrt{k}\log(kr)}{n^2} \right) \cdot \|B \odot C\|_F \|A\| \quad \text{by Lemma C.10)} \\
&\leq \left( \frac{c\sqrt{k}\log(kr)}{n^2} \right) \cdot \|A\|\|B\|\|C\|_F,
\end{aligned}$$

by Claim A.3. We have a similar bound for $\|\widehat{Y}\|_F$. Hence, we have

$$\begin{aligned}
\|\widetilde{\Phi} - \widehat{X} \odot \widehat{Y}\| &= \|(\widehat{X} + E_X) \odot (\widehat{Y} + E_Y) - \widehat{X} \odot \widehat{Y}\| \leq \|E_X \odot \widehat{X}\| + \|\widehat{Y} \odot E_Y\| + \|E_X \odot E_Y\| \\
&\leq \|E_X\|_F \|\widehat{Y}\| + \|E_Y\|_F \|\widehat{X}\| + \|E_X\|_F \|E_Y\|_F \quad \text{(using Claim A.3)} \\
&\leq \frac{c^2 k \log(kr)^2 \cdot \sqrt{\mathrm{OPT}}}{n^4} \left( 2\|A\| \cdot \|B\| \cdot \|C\|_F + \sqrt{\mathrm{OPT}} \right).
\end{aligned}$$

$\square$

We now finish the proof of Theorem B.3.

**Proof of Theorem B.3** Our analysis will work with any algorithm that computes the pseudoinverse solution. For any $M$, we find $\widehat{M}$ such that

$$\|\widehat{M} - M^\dagger\| \leq \varepsilon \sigma_{\min}(M)$$

Without loss of generality, we can assume that $\|T\|_F = 1$ (for scaling), and $\|A\|_F = \|B\|_F = \|C\|_F$ (since we can redistribute the mass among the factors of any decomposition arbitrarily). Let us denote by $\mathrm{OPT} = \|Err\|_F^2$, and $\kappa = \max\{\kappa(A), \kappa(B), \kappa(C)\}$. By Claim A.4, we have that $1 \leq \|A\|_F^3 \leq r\kappa^2 \leq r^{1+2c_0}$. For our purposes, we can think of $\mathrm{OPT} \geq \varepsilon$, since the guarantee and proof works up any upper bound on $\|Err\|_F^2$. The algorithm in every iteration solves a least squares problem up to precision $\varepsilon$ in time polynomial in $n, d, \log(1/\varepsilon)$. We can ignore this $\varepsilon$ in this robust analysis, since it is dominated by the error terms in the tensor, and the intermediate steps. Also note that $\mathrm{OPT} \leq 1$; in fact, it will be useful to think of $\mathrm{OPT} < 1/\mathrm{poly}(k, r)$, since otherwise the trivial bound suffices.

We can bound the objective value in the second iteration using Lemma C.9 and combine it with Lemma C.11 to get

$$\begin{aligned}
\left\| T - \sum_{i=1}^{k} x_i^{(1)} \otimes y_i^{(1)} \otimes z_i^{(2)} \right\|_F^2 &\leq 2(\mathrm{OPT} + \varepsilon) + \frac{O(1) \cdot \|\Phi - \widetilde{\Phi}\|^2 \cdot \|T\|_F^2}{\sigma_{r^2}(\Phi)^2} \\
&\leq O(\mathrm{OPT}) \left( 1 + \frac{O(k^2 \log^4(rk))}{n^8 \sigma_{r^2}(\Phi)^2} \left( \|A\|^2 \|B\|^2 \|C\|_F^2 + \mathrm{OPT} \right) \right) \\
&\leq O(\mathrm{OPT}) \left( 1 + \frac{O(k^2 r^2 \kappa^4 \log^4(rk))}{n^8 \sigma_{r^2}(\Phi)^2} \right) \\
&\leq O(\mathrm{OPT}) \cdot \mathrm{poly}(k, r),
\end{aligned}$$

by using the bound of $\sigma_{r^2}(\Phi) \geq (n^4 \mathrm{poly}(k, r))^{-1}$ from Theorem B.2, and the bound on $\kappa$ in the assumptions on Theorem B.3. This concludes the proof.

# D    Experimental Evaluation

Our theoretical results guarantee convergence of Algorithm 1 when the overparameterization $k = O(r^2)$. In our experiments we investigate whether this overparameterization factor is also observed in practice, and what the leading constant in the dependence is. The second question is how much overparameterization is needed i.e., does ALS require $k = \Omega(r^2)$ to succeed? Our experiments suggests that both these questions are true.

Algorithm 1 is a non-standard version of the ALS algorithm because in each iteration, it performs the updates to each mode in parallel. That is, $X^{(t+1)}, Y^{(t+1)}, Z^{(t+1)}$ are all a function of $X^{(t)}, Y^{(t)}, Z^{(t)}$. This is in contrast to the standard version of ALS which updates the modes sequentially. That is, $X^{(t+1)}$ will depend on $Y^{(t)}, Z^{(t)}$ as in the parallel version, but then $Y^{(t+1)}$ will depend on $X^{(t+1)}, Z^{(t)}$, and $Z^{(t+1)}$ will depend on $X^{(t+1)}, Y^{(t+1)}$. Our theoretical results focus on the parallel update version (Algorithm 1) because it is easier to analyze. In our experiments, we evaluate both the parallel-update and the standard sequential versions of ALS, to see what the effect of overparameterization is.

## D.1    Experimental setup

To evaluate the parallel-update version of ALS (Algorithm 1), we implemented a non-optimized version using the `scipy` least squares solver. For each $n, r, k$ that we analyze, we run 20 trials. For each trial we generate 3 random $n \times r$ factor matrices (each entry is an independent Gaussian) to make up our ground truth tensor. We initialize the factors of our model to be fully random $n \times k$ matrices. For all $n = 200$ and $n = 500$ we set the maximum number of iterations to be 20, due to computational constraints.

To evaluate standard ALS, we used the `parafac` method from the TensorLy library [KPAP19], which provides an optimized version of the standard (sequential) ALS method. As for parallel-update ALS, for each $n, r, k$ that we analyze we run 20 trials, and for each trial we generate 3 random (Gaussian) $n \times r$ factor matrices to make up our ground truth tensor. We then call `parafac` on this tensor with random initialization.[7] For $n = 500$ we set the maximum number of iterations to be 100, and for $n = 1000$ we set the maximum number of iterations to be 20, due to computational constraints.

We provide python code to run both experimental setups as part of the supplementary material.

## D.2    Parallel-updated ALS discussion

$k = r^2$ **suffices.**    Our theoretical results guarantee that parallel-update ALS (Algorithm 1) should converge in $O(1)$ steps as long as $k = \Omega(r^2)$. Our experiments validate that this holds for $k = r^2$, with no leading constant. In Figure 1, we plot the errors of running Algorithm 1 for $n = 200$, $r \in \{8, 11, 14, 17, 20\}$, and various values of $k$ that depend on the setting of $r$. We see that consistently across all settings of $r$, Algorithm 1 consistently fails to converge for any value of $k < r^2$ and consistently converges for $k \geq r^2$.

Our theoretical results guarantee that, once $n$ is sufficiently larger than $k$, the overparameterized rank necessary for parallel-update ALS (Algorithm 1) to succeed has no dependence on $n$. In Figure 2 we plot the errors of running Algorithm 1 for $r = 8$, and $n \in \{200, 500\}$. We observe that there is indeed no apparent difference in the results.

## D.3    Standard ALS discussion

$k \leq r^2$ **suffices for standard ALS.**    While our theoretical results apply to the parallel-update version of ALS, we observe that overparameterization $k = r^2$ seems to suffice to ensure convergence for the standard sequential version of ALS as well. We run this experiment for $n = 500$ and various values of $r$ and $k$, and the results can be found in Figure 3, Figure 4, Figure 5, Figure 6, Figure 7, Figure 8, Figure 9. In all of these experiments, we see that ALS starts to converge for values of $k$ that are less than $r^2$. Our theoretical result for the parallel-update ALS guarantees that the overparameterization necessary to ensure convergence should have no dependence on $n$. To evaluate whether this is true for

---

[7]We use fully random initialization as opposed to the default SVD initialization, which deviates significantly from what we analyze in this work.

standard ALS, we provide Figure 9 and Figure 10, which both evaluate $r = 20$ and the same values of $k$ for two different values of $n$ ($n = 500$ and $n = 1000$). We observe that the different choices of $n$ do not appear to have any significant impact on the error of standard ALS as a function of the overparameterization.

**Comparison to parallel-update ALS.**   We see that in comparison to the parallel-update version of ALS, the standard version has a more graceful degradation of error as a function of $k$. While we do not have a theoretical result that proves that standard ALS performs only better than the parallel-update version we analyze, it does appear in our experiments that this is the case. Even though standard ALS converges for smaller values of $k$ than the parallel-update ALS, we note that many of our experiments, including Figure 7, Figure 8, Figure 9, and Figure 10, seem to display that standard ALS experiences instability at values of $k$ very close to $r^2$, that it does not experience for other nearby values of $k$. We view this as an interesting phenomenon to investigate in future work.

**Necessary overparameterization.**   Our theoretical results guarantee that parallel-update ALS converges for $k = \Omega(r^2)$. Our experimental results suggest that parallel-update ALS converges exactly when $k \geq r^2$ (with no leading constant). Our experiments also suggest that standard (sequential) ALS converges for values of $k < r^2$. However, we observe that even though the input tensors in our experiment are chosen randomly from a nicely-behaved distribution , standard ALS still requires $k$ significantly larger than $r$ to converge. Our results are inconclusive as to what dependency $k$ must have on $r$ to ensure convergence. We view it as an exciting future direction of both theoretical and experimental work to understand the overparameterization necessary to ensure convergence of standard ALS.

### D.4   Data

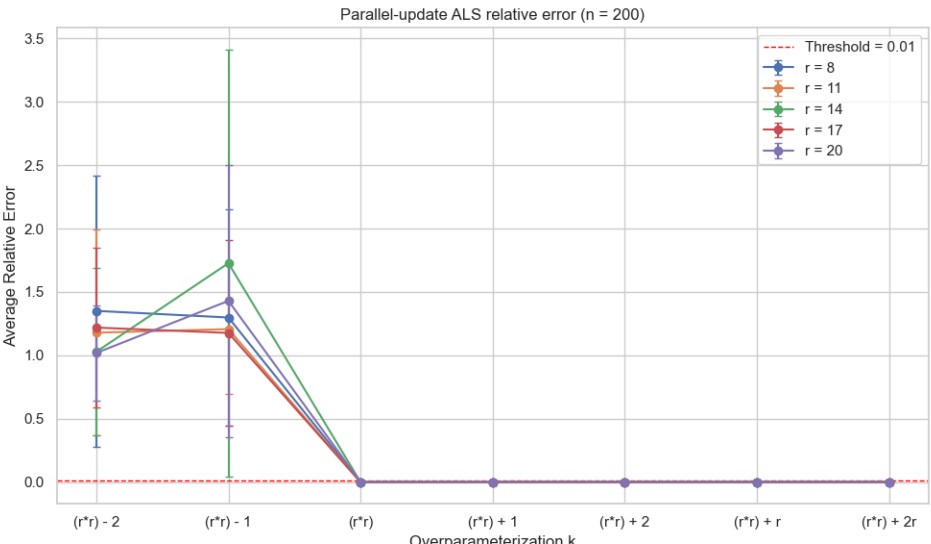

Figure 1: Results of running the parallel-update version of ALS (Algorithm 1) for $n = 200$, various values of $r$, and various values of $k$ that depend on $r$. We see that this method consistently fails to converge for $k < r^2$ and consistently converges for $k \geq r^2$. For this experiment we run ALS for a maximum of 20 iterations per trial. For trials where the method converged, it always converged in 2 iterations, which is consistent with our theoretical result. The reported values are aggregated over 20 independent trials, with error bars corresponding to one standard deviation. The data for this plot can be found in Figure 11.

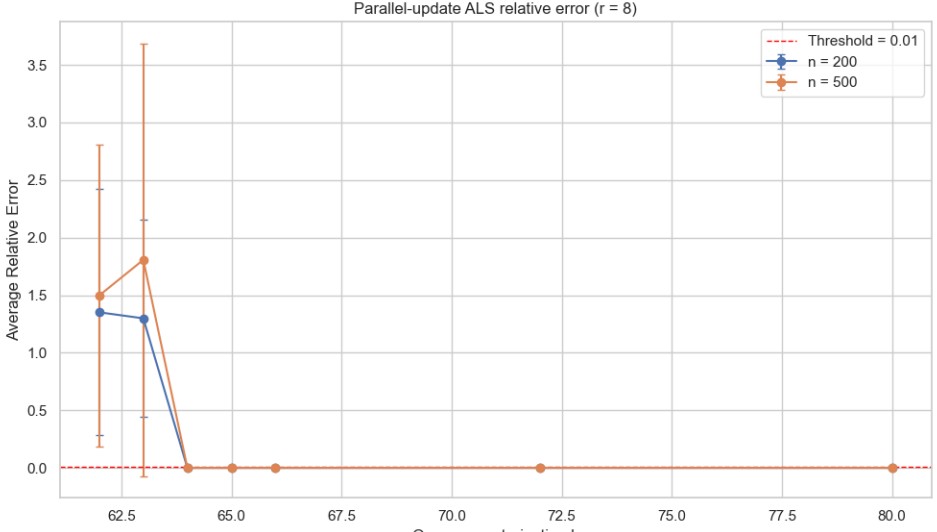

Figure 2: Results of running the parallel-update version of ALS (Algorithm 1) for $r = 8$, two values of $n$, and various values of $k$ that depend on $r$. We see that this method consistently fails to converge for $k < r^2$ and consistently converges for $k \geq r^2$. For this experiment we run ALS for a maximum of 20 iterations per trial. For trials where the method converged, it always converged in 2 iterations, which is consistent with our theoretical result. The reported values are aggregated over 20 independent trials, with error bars corresponding to one standard deviation. The data for this plot can be found in Figure 12.

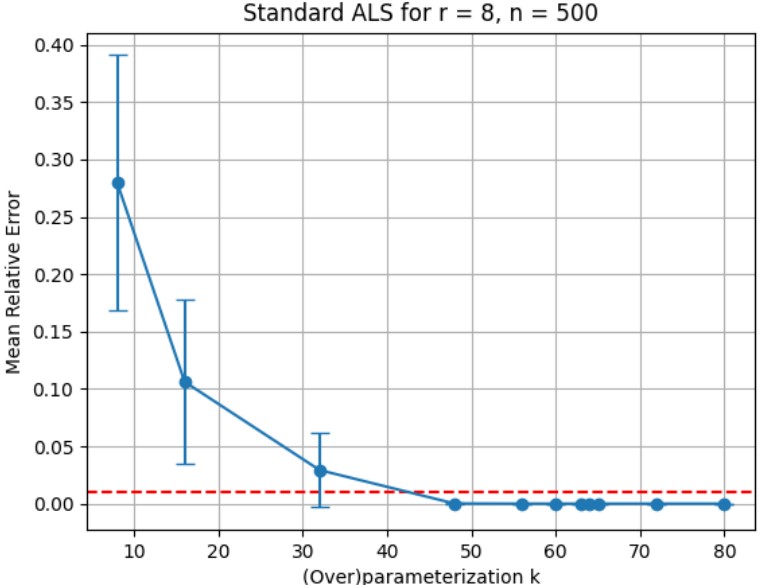

Figure 3: Results of running standard ALS (`parafac` by TensorLy [KPAP19]) for $n = 500$, $r = 8$ and various values of $k$. We observe that the error degrades gracefully as a function of $k$. The minimum $k$ necessary to ensure convergence seems to be significantly larger than $r = 8$, but smaller than $r^2 = 64$. For this experiment we run ALS for a maximum of 100 iterations per trial. The reported values are aggregated over 20 independent trials, with error bars corresponding to one standard deviation. The data for this plot can be found in Figure 13.

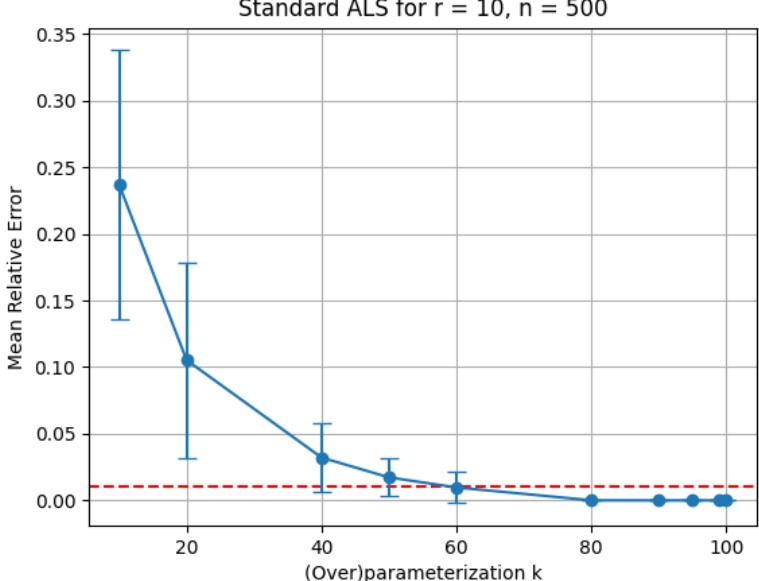

Figure 4: Results of running standard ALS (`parafac` by TensorLy [KPAP19]) for $n = 500$, $r = 10$ and various values of $k$. We observe that the error degrades gracefully as a function of $k$. The minimum $k$ necessary to ensure convergence seems to be significantly larger than $r = 10$, but smaller than $r^2 = 100$. For this experiment we run ALS for a maximum of 100 iterations per trial. The reported values are aggregated over 20 independent trials, with error bars corresponding to one standard deviation. The data for this plot can be found in Figure 14.

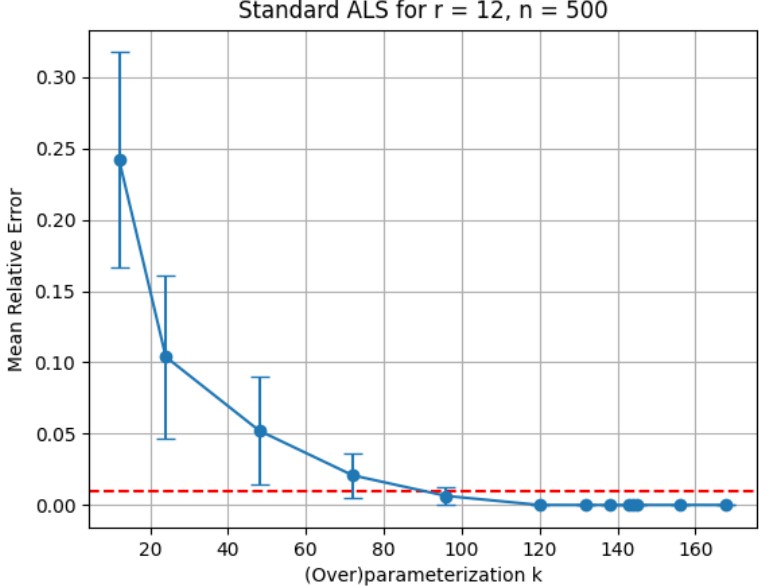

Figure 5: Results of running standard ALS (`parafac` by TensorLy [KPAP19]) for $n = 500$, $r = 12$ and various values of $k$. We observe that the error degrades gracefully as a function of $k$. The minimum $k$ necessary to ensure convergence seems to be significantly larger than $r = 12$, but smaller than $r^2 = 144$. For this experiment we run ALS for a maximum of 100 iterations per trial. The reported values are aggregated over 20 independent trials, with error bars corresponding to one standard deviation. The data for this plot can be found in Figure 15.

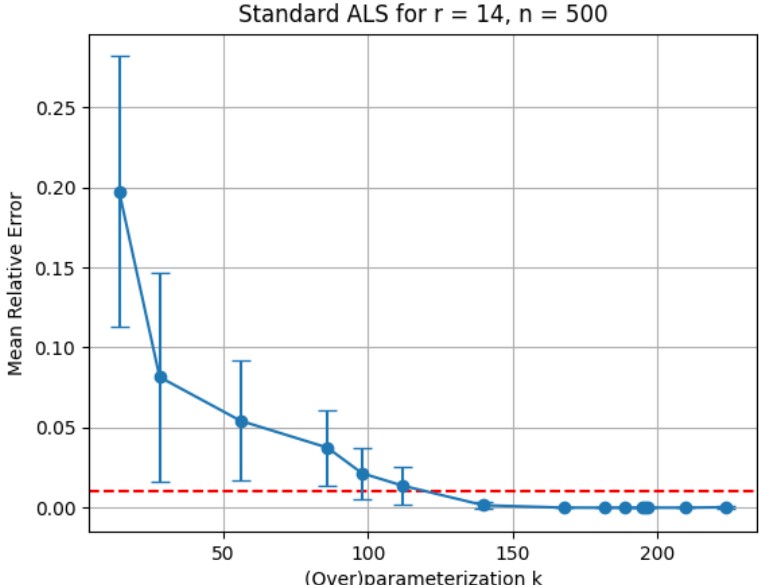

Figure 6: Results of running standard ALS (`parafac` by TensorLy [KPAP19]) for $n = 500$, $r = 14$ and various values of $k$. We observe that the error degrades gracefully as a function of $k$. The minimum $k$ necessary to ensure convergence seems to be significantly larger than $r = 14$, but smaller than $r^2 = 196$. For this experiment we run ALS for a maximum of 100 iterations per trial. The reported values are aggregated over 20 independent trials, with error bars corresponding to one standard deviation. The data for this plot can be found in Figure 16.

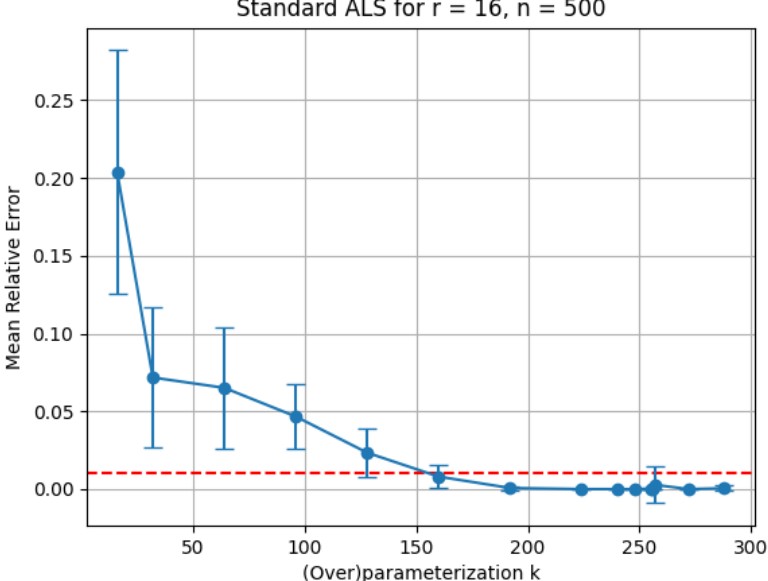

Figure 7: Results of running standard ALS (`parafac` by TensorLy [KPAP19]) for $n = 500$, $r = 16$ and various values of $k$. We observe that the error degrades gracefully as a function of $k$. The minimum $k$ necessary to ensure convergence seems to be significantly larger than $r = 16$, but smaller than $r^2 = 256$. We also observe that standard ALS seems to experience some instability for values of $k$ very close to $r^2 = 256$. For this experiment we run ALS for a maximum of 100 iterations per trial. The reported values are aggregated over 20 independent trials, with error bars corresponding to one standard deviation. The data for this plot can be found in Figure 13.

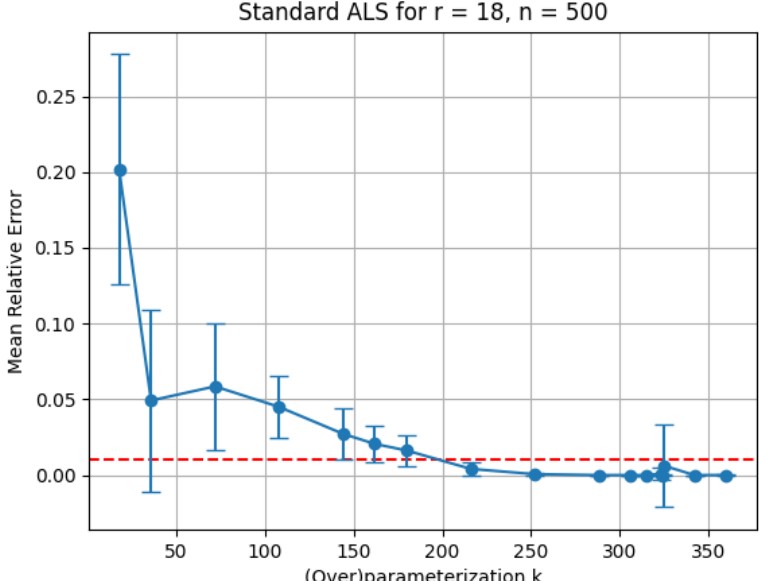

Figure 8: Results of running standard ALS (`parafac` by TensorLy [KPAP19]) for $n = 500$, $r = 18$ and various values of $k$. We observe that the error degrades gracefully as a function of $k$. The minimum $k$ necessary to ensure convergence seems to be significantly larger than $r = 18$, but smaller than $r^2 = 324$. We also observe that standard ALS seems to experience some instability for values of $k$ very close to $r^2 = 324$. For this experiment we run ALS for a maximum of 100 iterations per trial. The reported values are aggregated over 20 independent trials, with error bars corresponding to one standard deviation. The data for this plot can be found in Figure 18.

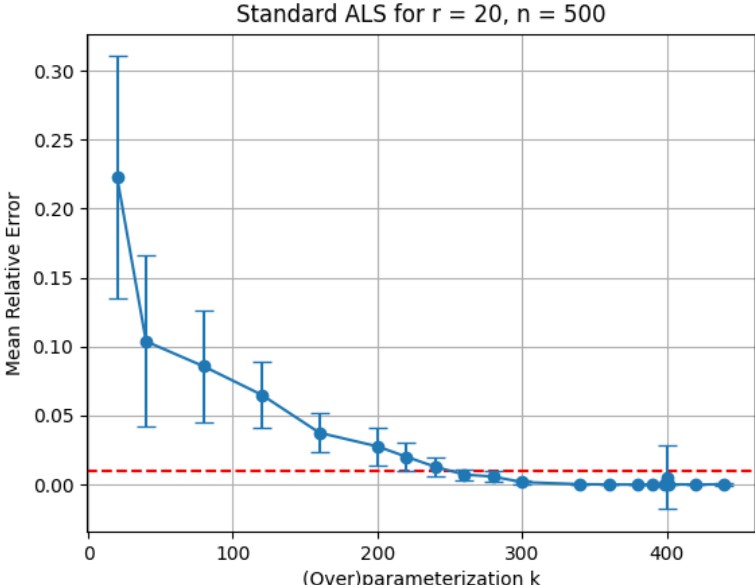

Figure 9: Results of running standard ALS (`parafac` by TensorLy [KPAP19]) for $n = 500$, $r = 20$ and various values of $k$. We observe that the error degrades gracefully as a function of $k$. The minimum $k$ necessary to ensure convergence seems to be significantly larger than $r = 20$, but smaller than $r^2 = 400$. We also observe that standard ALS seems to experience some instability for values of $k$ very close to $r^2 = 400$. For this experiment we run ALS for a maximum of 100 iterations per trial. The reported values are aggregated over 20 independent trials, with error bars corresponding to one standard deviation. The data for this plot can be found in Figure 19.

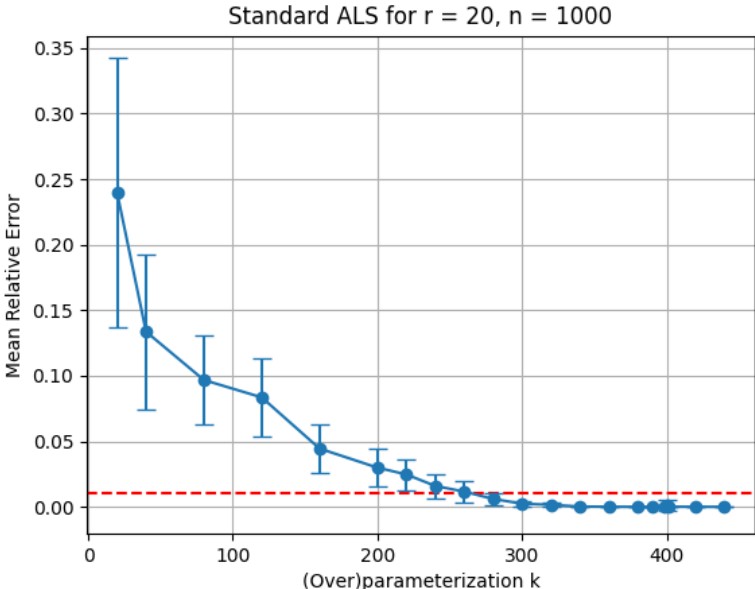

Figure 10: Results of running standard ALS (`parafac` by TensorLy [KPAP19]) for $n = 1000$, $r = 20$ and various values of $k$. We observe the results of this experiment are very similar to Figure 9, suggesting that convergence of standard ALS is a function of $k$ and not $n$. For this experiment we run ALS for a maximum of 100 iterations per trial. The reported values are aggregated over 20 independent trials, with error bars corresponding to one standard deviation. The data for this plot can be found in Figure 19.

| $r$ | $k$ | mean | std. dev |
|---|---|---|---|
| 8 | $r^2 - 2$ | 1.35 | 1.07 |
| 8 | $r^2 - 1$ | 1.3 | 0.855 |
| 8 | $r^2$ | 5.99e-13 | 9.46e-13 |
| 8 | $r^2 + 1$ | 3.21e-14 | 1.85e-14 |
| 8 | $r^2 + 2$ | 2.86e-14 | 1.89e-14 |
| 8 | $r^2 + r$ | 4.92e-14 | 1.66e-14 |
| 8 | $r^2 + 2r$ | 3.09e-14 | 4.6e-15 |
| 11 | $r^2 - 2$ | 1.18 | 0.813 |
| 11 | $r^2 - 1$ | 1.21 | 0.511 |
| 11 | $r^2$ | 3.38e-13 | 8.14e-13 |
| 11 | $r^2 + 1$ | 4.66e-14 | 4.06e-14 |
| 11 | $r^2 + 2$ | 2.81e-14 | 1.21e-14 |
| 11 | $r^2 + r$ | 1.25e-14 | 2.1e-15 |
| 11 | $r^2 + 2r$ | 9.46e-15 | 1.23e-15 |
| 14 | $r^2 - 2$ | 1.03 | 0.659 |
| 14 | $r^2 - 1$ | 1.73 | 1.69 |
| 14 | $r^2$ | 9.44e-13 | 3.14e-12 |
| 14 | $r^2 + 1$ | 6.93e-14 | 1.03e-13 |
| 14 | $r^2 + 2$ | 3.96e-14 | 1.49e-14 |
| 14 | $r^2 + r$ | 1.54e-14 | 1.53e-15 |
| 14 | $r^2 + 2r$ | 3.67e-14 | 4.32e-15 |
| 17 | $r^2 - 2$ | 1.22 | 0.628 |
| 17 | $r^2 - 1$ | 1.18 | 0.731 |
| 17 | $r^2$ | 2.55e-13 | 5.84e-13 |
| 17 | $r^2 + 1$ | 5.83e-14 | 3.82e-14 |
| 17 | $r^2 + 2$ | 4.44e-14 | 1.36e-14 |
| 17 | $r^2 + r$ | 1.56e-14 | 1.29e-15 |
| 17 | $r^2 + 2r$ | 9.45e-15 | 6.06e-16 |
| 20 | $r^2 - 2$ | 1.02 | 0.376 |
| 20 | $r^2 - 1$ | 1.43 | 1.07 |
| 20 | $r^2$ | 9.33e-13 | 1.04e-12 |
| 20 | $r^2 + 1$ | 8.34e-14 | 3.87e-14 |
| 20 | $r^2 + 2$ | 7.58e-14 | 2.77e-14 |
| 20 | $r^2 + r$ | 1.49e-14 | 8.28e-16 |
| 20 | $r^2 + 2r$ | 3.83e-14 | 3.45e-15 |

Figure 11: Data used to generate Figure 1. For these experiments $n = 200$, and the maximum number of iterations of ALS is 20. The reported values are aggregated over 20 independent trials.

| $n$ | $r$ | $k$ | mean | std. dev |
|-----|-----|-----|------|----------|
| 200 | 8 | $r^2 - 2$ | 1.35 | 1.07 |
| 200 | 8 | $r^2 - 1$ | 1.3 | 0.855 |
| 200 | 8 | $r^2$ | 5.99e-13 | 9.46e-13 |
| 200 | 8 | $r^2 + 1$ | 3.21e-14 | 1.85e-14 |
| 200 | 8 | $r^2 + 2$ | 2.86e-14 | 1.89e-14 |
| 200 | 8 | $r^2 + r$ | 4.92e-14 | 1.66e-14 |
| 200 | 8 | $r^2 + 2r$ | 3.09e-14 | 4.6e-15 |
| 500 | 8 | $r^2 - 2$ | 1.5 | 1.31 |
| 500 | 8 | $r^2 - 1$ | 1.81 | 1.88 |
| 500 | 8 | $r^2$ | 2.67e-12 | 1.11e-11 |
| 500 | 8 | $r^2 + 1$ | 3.36e-14 | 1.91e-14 |
| 500 | 8 | $r^2 + 2$ | 2.72e-14 | 1.25e-14 |
| 500 | 8 | $r^2 + r$ | 1.11e-14 | 2.17e-15 |
| 500 | 8 | $r^2 + 2r$ | 6.7e-15 | 1.29e-15 |

Figure 12: Data used to generate Figure 2. For these experiments the maximum number of iterations of ALS is 20. The reported values are aggregated over 20 independent trials.

| $r$ | $k$ | mean | std. dev. |
|-----|-----|------|-----------|
| 8 | 8 | 0.279915 | 0.111048 |
| 8 | 16 | 0.106255 | 0.071116 |
| 8 | 32 | 0.0293 | 0.032489 |
| 8 | 48 | 0.00011 | 0.000177 |
| 8 | 56 | 0.0 | 0.0 |
| 8 | 60 | 0.0 | 0.0 |
| 8 | 63 | 0.0 | 0.0 |
| 8 | 64 | 0.0 | 0.0 |
| 8 | 65 | 0.0 | 0.0 |
| 8 | 72 | 0.0 | 0.0 |
| 8 | 80 | 5e-06 | 2.2e-05 |

Figure 13: Data used to generate Figure 3. For this experiment $n = 500$, and the maximum number of iterations of ALS was 100. The reported values are aggregated over 20 independent trials.

| $r$ | $k$ | mean | std. dev. |
|-----|-----|------|-----------|
| 10 | 10 | 0.237 | 0.100775 |
| 10 | 20 | 0.105275 | 0.073401 |
| 10 | 40 | 0.03214 | 0.025882 |
| 10 | 50 | 0.01725 | 0.014379 |
| 10 | 60 | 0.009715 | 0.01181 |
| 10 | 80 | 5e-05 | 6.1e-05 |
| 10 | 90 | 0.0 | 0.0 |
| 10 | 95 | 0.0 | 0.0 |
| 10 | 99 | 3.5e-05 | 0.000157 |
| 10 | 100 | 4.6e-05 | 0.000113 |

Figure 14: Data used to generate Figure 4. For this experiment $n = 500$, and the maximum number of iterations of ALS was 100. The reported values are aggregated over 20 independent trials.

| $r$ | $k$ | mean | std. dev. |
| --- | --- | --- | --- |
| 12 | 12 | 0.241995 | 0.075478 |
| 12 | 24 | 0.10385 | 0.057524 |
| 12 | 48 | 0.052115 | 0.037644 |
| 12 | 72 | 0.020825 | 0.015514 |
| 12 | 96 | 0.006375 | 0.006497 |
| 12 | 120 | 1.5e-05 | 3.7e-05 |
| 12 | 132 | 0.0 | 0.0 |
| 12 | 138 | 0.0 | 0.0 |
| 12 | 143 | 0.0 | 0.0 |
| 12 | 144 | 0.0 | 0.0 |
| 12 | 145 | 0.0 | 0.0 |
| 12 | 156 | 2e-05 | 8.9e-05 |
| 12 | 168 | 0.0 | 0.0 |

Figure 15: Data used to generate Figure 5. For this experiment $n = 500$, and the maximum number of iterations of ALS was 100. The reported values are aggregated over 20 independent trials.

| $r$ | $k$ | mean | std. dev. |
| --- | --- | --- | --- |
| 14 | 14 | 0.1976 | 0.08427 |
| 14 | 28 | 0.081565 | 0.065547 |
| 14 | 56 | 0.054265 | 0.03741 |
| 14 | 86 | 0.037515 | 0.023547 |
| 14 | 98 | 0.02151 | 0.015892 |
| 14 | 112 | 0.01377 | 0.011933 |
| 14 | 140 | 0.001525 | 0.002253 |
| 14 | 168 | 1e-05 | 3.1e-05 |
| 14 | 182 | 0.0 | 0.0 |
| 14 | 189 | 0.0 | 0.0 |
| 14 | 195 | 0.0 | 0.0 |
| 14 | 196 | 5e-06 | 2.2e-05 |
| 14 | 197 | 1e-05 | 3.1e-05 |
| 14 | 210 | 0.0 | 0.0 |
| 14 | 224 | 0.00027 | 0.001207 |

Figure 16: Data used to generate Figure 6. For this experiment $n = 500$, and the maximum number of iterations of ALS was 100. The reported values are aggregated over 20 independent trials.

| $r$ | $k$ | mean | std. dev. |
|-----|-----|------|-----------|
| 16 | 16 | 0.20394 | 0.078098 |
| 16 | 32 | 0.07173 | 0.045191 |
| 16 | 64 | 0.065095 | 0.039075 |
| 16 | 96 | 0.04665 | 0.02052 |
| 16 | 128 | 0.023405 | 0.015258 |
| 16 | 160 | 0.008025 | 0.007481 |
| 16 | 192 | 0.000725 | 0.001254 |
| 16 | 224 | 5e-06 | 2.2e-05 |
| 16 | 240 | 0.0 | 0.0 |
| 16 | 248 | 0.0 | 0.0 |
| 16 | 255 | 0.00012 | 0.000537 |
| 16 | 256 | 3.5e-05 | 7.5e-05 |
| 16 | 257 | 0.00272 | 0.011766 |
| 16 | 272 | 5e-06 | 2.2e-05 |
| 16 | 288 | 0.000575 | 0.001772 |

Figure 17: Data used to generate Figure 7. For this experiment $n = 500$, and the maximum number of iterations of ALS was $100$. The reported values are aggregated over 20 independent trials.

| $r$ | $k$ | mean | std. dev. |
|-----|-----|------|-----------|
| 18 | 18 | 0.201895 | 0.075775 |
| 18 | 36 | 0.04922 | 0.060002 |
| 18 | 72 | 0.058525 | 0.041778 |
| 18 | 108 | 0.045035 | 0.020357 |
| 18 | 144 | 0.02728 | 0.016545 |
| 18 | 162 | 0.02057 | 0.011974 |
| 18 | 180 | 0.01624 | 0.009968 |
| 18 | 216 | 0.004095 | 0.004536 |
| 18 | 252 | 0.000635 | 0.001181 |
| 18 | 288 | 5e-06 | 2.2e-05 |
| 18 | 306 | 0.0 | 0.0 |
| 18 | 315 | 0.0 | 0.0 |
| 18 | 323 | 0.000965 | 0.004269 |
| 18 | 324 | 3.5e-05 | 6.7e-05 |
| 18 | 325 | 0.00613 | 0.027297 |
| 18 | 342 | 5e-06 | 2.2e-05 |
| 18 | 360 | 3e-05 | 0.000134 |

Figure 18: Data used to generate Figure 8. For this experiment $n = 500$, and the maximum number of iterations of ALS was $100$. The reported values are aggregated over 20 independent trials.

| $r$ | $k$ | mean | std. dev. |
|---|---|---|---|
| 20 | 20 | 0.222785 | 0.087655 |
| 20 | 40 | 0.103795 | 0.062048 |
| 20 | 80 | 0.085645 | 0.040937 |
| 20 | 120 | 0.06505 | 0.023917 |
| 20 | 160 | 0.037505 | 0.014094 |
| 20 | 200 | 0.027705 | 0.013622 |
| 20 | 220 | 0.02015 | 0.009954 |
| 20 | 240 | 0.0129 | 0.007192 |
| 20 | 260 | 0.00727 | 0.004 |
| 20 | 280 | 0.005775 | 0.003874 |
| 20 | 300 | 0.001783 | 0.001685 |
| 20 | 340 | 0.000275 | 0.000197 |
| 20 | 360 | 4.5e-05 | 5.1e-05 |
| 20 | 380 | 0.0 | 0.0 |
| 20 | 390 | 0.0 | 0.0 |
| 20 | 399 | 6.5e-05 | 0.000208 |
| 20 | 400 | 0.005205 | 0.023042 |
| 20 | 401 | 0.000145 | 0.000417 |
| 20 | 420 | 0.0 | 0.0 |
| 20 | 440 | 0.00019 | 0.00085 |

Figure 19: Data used to generate Figure 9. For this experiment $n = 500$, and the maximum number of iterations of ALS was 100. The reported values are aggregated over 20 independent trials.

| $r$ | $k$ | mean | std. dev. |
|---|---|---|---|
| 20 | 20 | 0.239725 | 0.102431 |
| 20 | 40 | 0.13363 | 0.059243 |
| 20 | 80 | 0.096775 | 0.033897 |
| 20 | 120 | 0.0835 | 0.029776 |
| 20 | 160 | 0.044465 | 0.01832 |
| 20 | 200 | 0.02991 | 0.01427 |
| 20 | 220 | 0.024735 | 0.011906 |
| 20 | 240 | 0.01598 | 0.00924 |
| 20 | 260 | 0.011645 | 0.008376 |
| 20 | 280 | 0.00604 | 0.004469 |
| 20 | 300 | 0.002445 | 0.002042 |
| 20 | 320 | 0.00157 | 0.001794 |
| 20 | 340 | 0.00025 | 0.000173 |
| 20 | 360 | 5e-05 | 5.1e-05 |
| 20 | 380 | 0.0 | 0.0 |
| 20 | 390 | 0.0 | 0.0 |
| 20 | 399 | 7.5e-05 | 0.000251 |
| 20 | 400 | 0.001155 | 0.004456 |
| 20 | 401 | 0.0001 | 0.000296 |
| 20 | 420 | 2e-05 | 8.9e-05 |
| 20 | 440 | 0.0 | 0.0 |

Figure 20: Data used to generate Figure 10. For this experiment $n = 1000$, and the maximum number of iterations of ALS was 20. The reported values are aggregated over 20 independent trials.

