# OpenReview forum: "Guarantees for Alternating Least Squares in Overparameterized Tensor Decompositions"
_NeurIPS.cc/2025/Conference — NeurIPS 2025 spotlight_

### Official Review · Reviewer_mBVm · 2025-07-01

**Clarity:** 3
**Significance:** 4
**Originality:** 3
**Rating:** 5
**Confidence:** 3

**Summary:**

This paper proves that alternating least squares (ALS), when initialized randomly and overparameterized to $k=O(r^2)$, achieves global convergence with high probability for decomposing and approximating a third-order tensor of rank $r\ll n$. The main contributions are: 1. ALS with moderate overparameterization ($k = O(r²)$) ensures global convergence; 2. theoretical guarantees apply to both tensor decomposition and low-rank approximation; 3. Novel analytical tools are developed to study convergence in overparameterized settings.

**Questions:**

1. What is the minimal achievable value of $k$ for your guarantees? Is overparameterization fundamentally necessary to ensure theoretical guarantee for ALS, or could alternative approaches work without overparametrization?
2. What key techniques enable the improvement from prior overparameterization bounds $(k=O(r^{7.5\log n}$) to your result ($k=O(r^2)$)? Could you elaborate on how this reduction is achieved and why $k=O(r^2)$ is the threshold for your analysis?
3. Does Theorem 2.2 require the error term $E$ and the tensor $T$ to be symmetric? Or does the result hold under weaker structural assumptions?
4. A key distinction between matrix and tensor rank is that the factors $A,B,C$ in tensor decomposition may be rank-deficient. Your analysis assumes full-rank factors with bounded condition numbers — does this restriction simplify the proof, or could the results be extended to rank-deficient cases?

**Ethical Concerns:**

["NO or VERY MINOR ethics concerns only"]

**Final Justification:**

I appreciate the theoretical contributions of the paper and continue to support it. My rating remains unchanged.

**Limitations:**

yes

**Paper Formatting Concerns:**

No concern.

**Quality:**

4

**Strengths And Weaknesses:**

Strengths:
1. Improved Global Convergence Guarantees: The work establishes that ALS with moderate overparameterization ($k=O(r^2)$) achieves global convergence, a significant improvement over prior results requiring much larger $k$.
2. Broad Applicability: Theoretical guarantees extend beyond exact tensor decomposition to the more general low-rank approximation problem.
3. Novel Analytical Techniques: The analysis introduces innovative tools, such as matrix anticoncentration arguments, to study convergence in overparameterized settings.

Weaknesses:
1. Justification for Overparameterization: The necessity and sufficiency of the $k=O(r^2)$ requirement in the proof remain unclear. A deeper discussion on whether this bound is tight or merely an artifact of the analysis would strengthen the work.
2. Restrictive Well-Conditioned Assumption: The assumption that factors $A,B,C$ are well-conditioned excludes many realistic tensors and significantly simplifies the theoretical analysis. Extending the results to ill-conditioned or rank-deficient cases would enhance practical relevance.

---

> ### Author Rebuttal · Authors · 2025-07-31
>
> We thank the reviewer for the positive feedback and kind comments on the novelty of our techniques
>
>
> In terms of overparameterization, our experiments (included in the appendix) suggest that $r^2$ overparameterization is indeed necessary for the parallel version of ALS that we analyze to succeed.  For standard ALS, our experiments indicate that some degree of overparameterization is definitely necessary (and maybe even $\Omega(r^2)$), though it is less clear to what degree.  Both of these experimental observations hold even in the setting when $A, B, C,$ are generated randomly.  We think it is a very interesting direction of future work to analyze what degree of overparameterization is necessary for ALS to succeed.  That is, we provide an upper bound for parallel ALS, can one prove a lower bound on the overparameterized rank?
>
> The overparameterized rank being $r^2$ comes from the key observation that the slices of the tensor lie in a Khatri-Rao space of $A \odot A$ of dimension r, which is a subset of the larger Kronecker space $A \otimes A$ of dimension $r^2$.  At a high level, the Khatri-Rao space is very sensitive to the basis (columns) of $A$, whereas the Kronecker space $A \otimes A$ only depends on the subspace spanned by the columns of $A$, and not the individual basis vectors.  Thus, it is somehow an easier space to recover.  The main work in our analysis is in showing that this $A \otimes A$ space is indeed recovered by the iterates.
>
> The question about the full rank (or condition number) assumption of $A, B$ and $C$ is a very interesting question. It seems quite likely that if we make extra assumptions (e.g., some gap between the significant singular values and small singular values), we may be able to leverage the robust guarantee for low-rank approximations to move the projection onto the small singular vector directions onto the error term. However, this does not seem to work in general.
>
> In the main body, we sketched the proof assuming that the tensor $T$ and error matrix $E$ are symmetric, for ease of notation and also to convey the intuition of the proof.  Our theorems and proofs in the appendix extend to the asymmetric setting.

---

> > ### Comment · Reviewer_mBVm · 2025-08-08
> >
> > The experimental discussion on overparameterization is convincing; I appreciate the clarification. My support of the paper stands.

---

### Official Review · Reviewer_JdA9 · 2025-07-02

**Clarity:** 3
**Significance:** 1
**Originality:** 2
**Rating:** 4
**Confidence:** 4

**Summary:**

ALS-lke method (which would be more correctly called Paralles Least Squares) to the problem of decomposing a 3-way tensor into CP format is considered. It is assumed that all elements of the original tensor are known and it is small enough to fit in the computer memory; also it is known that the tensor is of $\epsilon$-$r$-CP-rank. The paper establishes that when decomposing into CP format of overparametrized rank $O(r^2)$ using the new method with random (Gaussain) initialisation of cores presented by the authors, one can achieve convergence guarantees with high probability and a given accuracy.

**Questions:**

- Could your proof method have been extended to regular ALS?
- Could your proof method have been extended to the case of rank $k<r^2$ (or $k<r^{d-1}$  in general)?
- Could your proof method have been (non-trivially) extended to the case of tensors of higher dimension?
- What is the practical conclusion of your algorithm, why a new method is needed if for $k=r^2$  overparametrized CP decomposition if the latter can be constructed trivially?
- Can you somehow round your resulting decomposition to get a decomposition with lower rank (but possibly higher error)?
- Why explore CPD when there are other tensor decompositions that are much better in the above context (obtaining almost exact rank), e.g. for Tensor-Train we can get an overparametrized decomposition, but then round it to the desired rank?

**Ethical Concerns:**

["NO or VERY MINOR ethics concerns only"]

**Final Justification:**

*   I recognize the **significant theoretical challenge** overcome: proving convergence guarantees (from random initialization, with high probability) for *any* iterative method applied to CP decomposition requiring only *polynomial* overparametrization (O(r²)) is a non-trivial result.
*   The paper is well-written, and the technical analysis appears sound for the specific algorithm and setting (3D tensors, rank O(r²)).



The effect and importance of this point are limited by these things:
1.  The sole focus on $d=3$ limits how it compares and can be used versus prior studies claiming results for general $d$.
2.  The use of a non-standard algorithm whose empirical work is shown to be worse than standard ALS calls into question the practical significance of the results.
3.  Unresolved questions about if the methods can be used for standard ALS, lower ranks, or higher dimensions.
4.  A lack of work on real uses (real data, completion, rounding) and not enough reason for focusing theoretical work on CP decomposition given its real-world limits and other choices available.

**Limitations:**

The limitations are listed in the paper and my review: only 3D tensors, only the new algorithm is considered, and the entire tensor is already set with its elements.

**Quality:**

1

**Strengths And Weaknesses:**

__Strength__

- The article is well written, easy to read, the main claims are there in the text, the technical details have gone to Appendix.
- All major assertions are rigorously proven.

__Weakness__

**New algorithm**
One of the main weaknesses -- the paper uses its own (rather than the standard ALS) method to construct the decomposition. In fact, if the given tensor $\mathcal T\in\mathbb R^{n_1\times n_2\times n_3}$ is known to be CP-rank $r$ (or $\epsilon$-$r$, i.e. can be represented with CPD of rank $r$ with accuracy $\epsilon$), then the construction of its CPD of rank $r^2$ is trivial. I will show that. For such a tensor all matricizations have rank $r$ (or $\epsilon$-rank), i.e. we can use (truncated)-SVD to obtain the decomposition of the first matricization: $T_1=$reshape($\mathcal T$, $n_1$, $n_2n_3$) = $U\Sigma_1V_1^T$, where $U\in\mathbb R^{n_1\times r}$. Similarly for the last matricization: $T_3=$reshape($\mathcal T$, $n_1n_2$, $n_3$) = $U_3\Sigma_3V^T$, where $V\in\mathbb R^{n_3\times r}$. Note the property of nullspace of $U$ and $V$: if $U^Tx=0=\sum_iU[i,j]x[i]$, then $\sum_iT[i,j,k]x[i]=0$, and similarly for $V$. Consider a 3-way core tensor $\mathcal G=\mathcal T\times_1U\times_3V=\sum_{i,j}\mathcal T[i,:,j]U[i,:]V[j,:]\in\mathbb R^{r\times n_2\times r}$. After swapping the dimensions and matricization of this tensor, we obtain the matrix $G=$reshape($\mathcal G'$, $n_2$, $r^2$). Now let $c_1=U\otimes 1_r\in\mathbb R^{n_1\times r^2}$, where $\otimes$ is the Kronecker product and  $1_r\in\mathbb R^{1\times r}$ is the vector of length $r$ containing ones. Similarly, let  $c_3=1_r \otimes V\in\mathbb R^{n_3\times r^2}$. Finally, remembering that matrices $U$ and $V$ have orthogonal columns: $U^TU=V^TV=I_r$ (and properties of their nullspaces), we obtain that for all admissible $i$, $j$, $k$, the following is true (or approximately true with accuracy $\epsilon$): $\mathcal T[i,j,k]=\sum_{\alpha=1}^{r^2}c_1[i,\alpha]G[j,\alpha]c_3[k,\alpha](=\sum_{\alpha_1}^r\sum_{\alpha_2}^rU[i,\alpha_1]\mathcal G[\alpha_1,j,\alpha_2]V[k,\alpha_2])$. Thus, we obtained an exact (in the case of exact ranks) decomposition of the original tensor into a CP format of rank $r^2$, and directly, without iterations. Moreover, this decomposition is obtained under milder conditions on the original tensor: it is sufficient that only two matricizations are of rank $r$, rather than the stricter condition in the paper at hand that the CPD $\mathcal T$ is of rank $r$, from which it follows, in particular, that all matricizations must be of rank no greater than $r$.

**Scalability**
In this paper tensors of dimension 3 considers only. For a tensor of arbitrary dimension $d$, which is known to have CPD of rank r, one can trivially (by the method described above) obtain the CP decomposition of rank $r^{d-1}$ directly. However, such tensors are not considered in the paper. Studying the proofs of the theorems led me to the idea that when extending the methods used by the authors to tensors of arbitrary dimension, the same estimate on the rank of the overparametrization is obtained (indeed, the authors talk about the total dimension of the columns of the cores of the decomposition except for the the one for which a local iteration is currently in progress, and since this dimension is a Kronekker product of the individual columns, it will be equal to $r^{d-1}$). In contrast, for example, the paper [1] (cited as SWZ19), which the authors try to generalise, studies tensors of common dimension (and obtains an estimate on the overparametrized rank $O((r/\epsilon)^{d-1})$ ). In the paper [2]  (cited as WWL+20) similarly, the result is obtained for the general case of d-way tensors (namely, the rank estimate is given as $O(r^{2.5d}\log n)$ for gradient descent algorithm).

**Practical relevance**
Thus, it is not very clear to me why one would want to study the properties of a new (iterative) algorithm that shows the same results as a naive direct algorithm. Furthermore, the authors restrict themselves to the case of 3-way tensors. Important cases where the dimensionality of the tensor is much larger, or the important problem formulation where we do not have all the values of the tensor but only a part of them (the so-called tensor completions problem), are not considered. The practical value of these results is questionable.

**Numerical Results.**
In Appendix, the authors present numerical experiments on synthetic data. It turns out that the standard ALS converges better than the presented algorithm. Moreover, it turns out that if the rank is at least 1 less than $r^2$, the presented new algorithm does not converge to almost zero error. It would be interesting to prove this behaviour in general, but it is not proved. On the contrary, the standard ALS behaves better even for overparametrized rank values smaller than $r^2$ (see figs. 3--10). And it is precisely this behaviour of ALS that is important to study from a practical point of view. Besides, it is important to know how ALS behaves not on absolutely random data (but, for example, on sparse data, or taken from real-world dataset). It is also important to study ALS convergence not on random cores initialisations, but, for example, with HOSVD. None of the above is mentioned in the article.

**Other tensor decomposions**
The motivation for choosing exactly the CP decomposition for the purposes in the paper is not clear. There are many other tensor decompositions. For example, for the Tensor-Train decomposition there is a method for its constructing (so-called TT-cross approximation method), which gives also overparametrized rank, and for which there are guarantees ([3]). In this case, the obtained overparametrized TT-decomposition can be later rounded off to a decomposition with a smaller rank [4].

[1]. Zhao Song, David P. Woodruff, and Peilin Zhong. Relative error tensor low rank approximation. In Symposium on Discrete Algorithms (SODA), 2019

[2]. Xiang Wang, Chenwei Wu, Jason D Lee, Tengyu Ma, and Rong Ge. Beyond lazy training for over-parameterized tensor decomposition. In H. Larochelle, M. Ranzato, R. Hadsell, M.F. Balcan, and H. Lin, editors, Advances in Neural Information Processing Systems, volume 33, pages 21934–21944. Curran Associates, Inc., 2020.

[3]  Stanislav Budzinskiy, Nikolai Zamarashkin. Tensor train completion: Local recovery guarantees via Riemannian optimization, 2023.

[4] I. V. Oseledets. Tensor-train decomposition. SIAM J. Sci. Comput., 33(5):2295–2317, 2011

Minor ans suggestions


- L129 ALS does not face some **fo** the earlier issue
- L261L "where in the first line $<\cdot>$ denotes the linear span, and in the second line $\otimes$" 1) The sentence is cut short 2) The notations has already been introduced
- L174--175 : flatten -> better use `reshape`
- L275 : (Algorithm 1, Line 8) Line **16**?
- L288 Gaussiansan
- L292 o(1). Usually O(1) is used to denote the constant (which does not depend on anything). o(1) implies that this value tends to zero. However, it is not clear at what limit it tends to zero. It is better to give more detailed asymptotics.
- It is better to use three separate characters for outer, Kronecker and Khatri-Rao products.
- Better use logscale on error plots

---

> ### Author Rebuttal · Authors · 2025-07-31
>
> We thank the reviewer for the detailed feedback. We would like to respectfully push back on the main motivation for this work. As we also mention in our paper, there are indeed other existing polynomial time algorithms that find decompositions of rank $r^2$. Most of these methods (including the one suggested by the reviewer) involve singular value decomposition (SVD) or variants to find the span of the factors. We remark that this is mentioned early in the paper; see lines 75-77 including references to BCV2014 and SWZ2019.
>
> Instead, the main motivation of our work is to analyze popular iterative heuristics for non-convex optimization. Tensor decomposition is seen as a canonical non-convex optimization problem, that is a stepping stone to more complex non-convex optimization problems that frequently arise in machine learning. This is exactly the  motivation behind the important work of Rong Ge, Jason Lee, Tengyu Ma, Xiang Wang, Chenwei Wu (published in NeurIPS 2020) that analyzes gradient descent for tensor decomposition; in fact they require an overparametrization of around $r^{7.5}$ which is much worse than the $r^2$ for more sophisticated SVD-based approaches (and even incur extra logarithmic factor in the ambient dimension n). In comparison our work analyzes a (parallel) variant of ALS, the most popular iterative heuristic for tensor decomposition; in general, this falls under the class of block-coordinate descent methods, another popular heuristic for optimization problems. We prove that a rank of $O(r^2)$ suffices (we are unaware of any such result for ALS-type methods). Admittedly we analyze a slight variant of the most popularly used ALS method but we hope that our techniques can be extended in future work to give theoretical guarantees for the standard ALS method. Providing theoretical guarantees for popular iterative methods involves significant technical challenges (over SVD-based methods) which we outline and overcome in this work. Note that the work of Ge and others also had to contend with many such challenges as well (and also analysed a variant of gradient descent). Such theoretical analyses of iterative heuristics are already interesting for tensor decomposition. But they become even more exciting from the broader perspective of non-convex optimization; we hope that they may reveal insights into the optimization landscape and that these ideas translate to other non-convex problems. I hope this conveys our main motivation, and why this  research direction is interesting;  this interest also seems to be reflected in the excitement of the other positive reviews.
>
> We agree that it would be interesting to extend our techniques to higher order tensors, and we leave this for future work. We suspect that these techniques should help; nevertheless decomposing order 3 tensors is already a computationally difficult non-convex problem and to the best of our knowledge few theoretical guarantees were shown for iterative methods even in this setting.

---

> ### Comment · Reviewer_JdA9 · 2025-08-03
>
> Thank you for your detailed response. I appreciate the authors' added clarity about why they analyzed iterative heuristics in non-convex optimization. I remain unconvinced that the paper warrants a much higher score. My concerns stem from specific limits in the study's scope and real-world use, which the response did not fully address.
>
> **Key Concerns:**
>
> 1.  **Limited Scope of Dimensions & Misleading Comparison to Previous Studies:**
>     *   The results in the paper are limited to 3-way tensors. The authors admit that expanding the analysis to higher dimensions ($d > 3$) is a complex task for the future.
>     *   This limit greatly lessens the perceived advance over prior studies. The authors state that WWL+20 needed overparametrization of $O(r^{7.5})$ for $d=3$ (compared to their $O(r^2)$), but WWL+20 had results for general d-way tensors. Likewise, SWZ19 offers a bound $O((r/ε)^{d-1})$ for general $d$. Thus, the paper's work is an *isolated result for $d=3$*, not a generalization. It is misleading to claim an advance over these studies without noting this major dimensional limit, as $d=3$ is often less useful than higher-dimensional settings in real uses. The exponential advance over WWL+20 for $d=3$ is noted, but its effect is lessened by this limit.
>
> 2.  **Algorithm Selection and Gap in Real-World Use :**
>     *   The response makes a solid point that studying iterative methods (like this version of ALS) for basic non-convex problems (like CP decomposition) is theoretically interesting per se, separate from giving the most practical decomposition method.
>  I acknowledge this motivation and the inherent theoretical challenge in proving convergence guarantees for iterative methods from random initialization. This distinction between algebraic existence and iterative convergence is a valid contribution.
>     *   Yet, considering a non-standard ALS (Parallel Least Squares) instead of the common used standard ALS is still a major weakness. The authors hope their methods can apply to standard ALS, but this is only a guess now. Also, the experiments show that standard ALS is better than the proposed method empirically, even for ranks below $r^2$. This makes a disconnect: the theoretical guarantees are for an algorithm that is empirically worse than a standard one on the very problem being studied. Showing that these guarantees affect how standard iterative methods work in practice is key.
>
> 3.  **Unaddressed Technical Scope and Real-World Worth:**
>     *   **Unanswered Questions:** Questions about expanding the proof method to standard ALS, to ranks $k < r^2$ (where standard ALS does better empirically), or to $d>3$ are still unanswered. The response focuses on rationale but does not handle these limits of the work now.
>     *   **Rationale for CP:** CP is a basic non-convex problem, but the response does not fully explain why it was chosen over other unstable formats or why theoretical guarantees for CP decomposition are so helpful, given its known practical issues (ill-posedness, sensitivity) and more dependable choices (like Tensor Train) with stable algorithms and convergence guarantees. The historical role is clear, but a better reason for its ongoing theoretical study in this context would help the paper.
>     *   **Limited Study of Real Uses:** The numerical experiments only use synthetic random data and compare it to ALS. There is no work with real-world data, sparse tensors, tensor completion (missing data), or the important step of rounding the overparametrized $O(r^2)$ decomposition to a lower rank -- a common thing done with other decompositions like TT. The practical worth and dependability of the method are unclear.
>
> **Overall:**
>
> The effect and importance of this point are limited by these things:
> 1.  The sole focus on $d=3$ limits how it compares and can be used versus prior studies claiming results for general $d$.
> 2.  The use of a non-standard algorithm whose empirical work is shown to be worse than standard ALS calls into question the practical significance of the results.
> 3.  Unresolved questions about if the methods can be used for standard ALS, lower ranks, or higher dimensions.
> 4.  A lack of work on real uses (real data, completion, rounding) and not enough reason for focusing theoretical work on CP decomposition given its real-world limits and other choices available.
>
> The theoretical analysis for the setting is valid and the reason for studying iterative methods is accepted, but these limits stop me from raising the score much. The work is a key but specific theoretical step; making its scope wider or showing clearer real-world use or if it can be used for standard algorithms would greatly raise its effect. Thus, I rise my score to 4 (Borderline accept).

---

> > ### Author Response · Authors · 2025-08-06
> >
> > Thank you for your detailed comments following our rebuttal. We very much appreciate your going through our rebuttal and updating your scores accordingly.
> >
> > We would be very happy to address your questions. We start with some of the more important ones:
> >
> > Rationale for CP: One of our main motivations for studying ALS and CP decomposition is understanding non-convex optimization (and this is the same for the paper of [Wang et al.] on “Beyond lazy-training…”).  There are formal connections between them. The paper of Ge-Lee-Ma 2018 (Theorem 2.1) and Awasthi-Tang-Vijayaraghavan 2021 (Prop 3.6) show a strong connection between tensor CP decompositions and training 2-layer neural networks. In particular, the least-squares objective for training 2-layer neural networks can be decomposed into a sum of CP-decomposition problems of increasing order. This is not true for other tensor decomposition problems. For this (and other reasons), tensor CP decompositions is extremely interesting from an ML standpoint, and is often seen as a stepping stone to understanding training of neural networks not just because of the non-convexity.
> >
> > We will respond to some of the other questions  in the next response.

---

> ### Author Response · Authors · 2025-08-06
>
> Regarding the limitation on $d=3$, we will make sure to emphasize that our comparison with the work of Ge et al is only valid for $d=3$ to avoid confusion. We believe that it is possible that a generalization of our techniques could give a bound of $O(r^{d-1})$ for higher order tensors. At a technical level, one would have to show a generalization of Theorem 4.1, namely that (for order 4 tensors for example, with matrix variables X,Y,Z and W) $\sigma_{r^3}(\hat{X}\odot \hat{Y}\odot \hat{Z})\geq \frac{1}{\mathrm{poly}(n,k)}$. It is quite likely that our arguments can be extended to show this, it would however require overcoming challenges due to limited independence of matrices $\hat{X}, \hat{Y}$ and $\hat{Z}$.  We would also like to note that the 3rd order case is itself interesting, and there are other works in the literature that focus entirely on order 3 tensors (e.g. [Sharan, Valiant 2017]: Orthogonalized ALS: A Theoretically Principled Tensor Decomposition Algorithm for Practical Use, [Ge Ma 2015]: Decomposing Overcomplete 3rd Order Tensors using Sum-of-Squares Algorithms)

---

### Official Review · Reviewer_sE4m · 2025-07-02

**Clarity:** 4
**Significance:** 4
**Originality:** 4
**Rating:** 5
**Confidence:** 4

**Summary:**

This paper provides a new analysis for a commonly used method, alternating least squares (ALS), for computing the CP decomposition of an order-3 tensor. Computing a decomposition with minimal rank can be computationally intractable. An alternative approach is to compute a decomposition with higher rank (but hopefully not too high). This paper shows that with high probability, under random initialization, targeting a rank $k = O(r^2)$ is enough, where $r$ is the actual CP rank.

**Questions:**

**Note:**  In general, my opinion is that a conference paper should be reviewable based on the main body of the paper given the limited time to do multiple reviews.

It is a great academic tradition to have questions which are more of a comment.

* I believe the author(s) should amend the title to reflect that the results hold only for CP decompositions. As it stands, the title is misleading since there are many kinds of tensor decompositions.

* The reference to Hillar and Lim (2013) is incorrect in the references: It appeared in JACM. While I did not check the entire bibliography, this kind of basic error suggests that the authors should check their references more carefully.

* I was surprised that there was no reference to Håstad's (famous) 1990 paper on the hardness of tensor rank over $\mathbb{Q}$. The Hillar and Lim paper contains a lot of discussion on the model of computation and relationship between rank over different fields.  There is comparatively little discussion here: perhaps a sentence to clarify would help.

* While improving the sufficient rank for admitting a decomposition, is there any hope for lower bounds? Perhaps this could be an interesting question for future work.

* The main technical result in the paper is in Theorem 4.1 and actual bound on the probability of the relevant event comes from the Carbery-Wright inequality. Since this is a result about a particular estimator and not a more general statement about singular values of Khatri-Rao products, is it possible to extract a lemma which could be of use elsewhere?

* The fact that any experimental evidence was left to the supplement and *not even mentioned in the main paper* is a red flag for me. Sneaking them "in the back door" through the supplement is, in my opinion, a violation of ethical norms. The overall research direction would surely benefit from careful experiments. However, simply running experiments on various flavors of ALS and not other methods (gradient descent, tensor power method, etc.) is not a proper experimental design. The authors should remove the experiments and leave practical evaluation to future work.

Some minor suggestions:

* (line 192) I found the "polynomial time" and "$O(1)$ steps" combination a bit confusing. What are the parameters of interest and which ones are going $\to \infty$ in these statements should be clarified. I understand this is coming from the black-box linear system solver but the statement could be clearer.
* Relatedly, please provide a standard reference for the solver you use.
* (p2) "mildly conditioned" is a little vague on a first reading. Perhaps make it a little more specific?
* Why state the result for the symmetric case and not the general case?
* (line 288) "Gaussianand" --> "Gaussian and"

**Ethical Concerns:**

["NO or VERY MINOR ethics concerns only"]

**Limitations:**

Yes.

**Paper Formatting Concerns:**

N/A.

**Quality:**

4

**Strengths And Weaknesses:**

**Strengths:**

* The new analysis opens the door to the analysis of ALS for other decompositions.
* The technical lemma may have uses in other applications involving optimization over tensor spaces.

**Weaknesses:**

* There is some gap between what the paper claims and what the results appear to be.
* The experiments are inadequate (but not needed).

---

> ### Author Rebuttal · Authors · 2025-07-31
>
> Thank you very much for your thoughtful feedback and comments. We will implement the changes that you proposed. We will also update the bibliography more carefully; thank you also for suggesting that we cite the work of Hastad.
>
>  For the experiments, our intent was not to claim that ALS is superior to other existing methods for tensor decompositions. As you pointed out, the main contribution of our work is the theoretical analysis under overparametrization; we hope these techniques also extend to other iterative methods under overparameterization. Hence we chose to focus solely on the theoretical portion for the main body of the paper. The main purpose of our experiments was to bring attention to what we think is a particularly interesting question about the extent of overparametrization needed for global convergence of ALS and its variants. This is directly related to one of the questions you posed about lower bounds on the rank of the decomposition. Our analysis suggests that $O(r^2)$ overparameterization suffices (we prove this for the parallel variant). Our experiments suggest that sufficient overparameterization (potentially even up to $\Omega(r^2)$ ) may be necessary as well (the experiments are most definitive for the parallel variant); proving such lower bounds seems quite challenging and interesting in its own right. We think this would be an interesting direction for future work. To your point, we do not intend our experiments to be taken as an evaluation of (either variant of) ALS as a method, and such evaluations require thought and care.  We appreciate you pointing this out, and we will make this point more clear in our revision.

---

### Official Review · Reviewer_L37m · 2025-07-03

**Clarity:** 1
**Significance:** 3
**Originality:** 3
**Rating:** 4
**Confidence:** 2

**Summary:**

This paper discusses the accuracy of alternating least squares for tensor decompositions under the overparameterized setup. For a third order tensor $n\times n \times n$ with CP rank $r$, this work shows that ALS overparameterized with rank $r$ achieves global convergence with high probability under random initialization.

**Questions:**

n/a

**Ethical Concerns:**

["NO or VERY MINOR ethics concerns only"]

**Final Justification:**

Raise the score as the rebuttal provided by the authors clear a couple of my concerns. This is a great contribution, but it would be good for authors to improve the clarity in the main text of the work.

**Quality:**

2

**Strengths And Weaknesses:**

Strengths: The introduced bound of $O(r^2)$ is much better than the previous work of $O(r^{7.5})$ and is impressive.

Weaknesses: One major weakness of this work is its presentation. As a theoretical study without experimental results included in the main text, readers will expect a more detailed algorithmic analysis and rigorous proofs. However, this work only provides a brief sketch of the analysis on pages 8 and 9, with many important analysis missing. Detailed comments are presented below:
1. lines 262-264: "suppose Y^t and Z^t were each drawn independently and randomly from colspan(A), with high probability $colspan(A\otimes A) \subseteq colspan(X^t\odot Y^t)$": this statement is nontrivial, it would be good to include the reference if this has been proved in the previous work, or provide a formal theorem with (sketched) proof.
2. Line 268: "Of course, the columns of X and Y are not initialized randomly in the span of A, instead they are random in the whole n-dimensional space." one important part that is missing is to show that with $k=\Omega(r^2)$, the subspace spanned by $A$ will be in the subspace of $X,Y,Z$ that are randomly generated with high probability.
3. Line 275: "The first step of ALS (Algorithm 1, Line 8) will set Z so that the second term is 0, which since X and Y are randomly initialized means setting $Z^{\perp}=0$": as this step is fixing X, Y and update Z with solving a least squares problem, maybe it's better to direct explain in the matrix format rather than keeping the CP format.
4. Line 283: "Thus, proving (4) reduces to showing that $X^1 \odot Y^1$ has rank $r^2$": this statement is nontrivial, it would be good to provide a formal theorem with proof.
5. It’s unclear how to derive the equation below 294, and why is it important.


Nit: in line 261, "where in the first line $\langle\cdot\rangle$ denotes the linear span, and in the second line $\odot$" should be "...,  in the second line $\odot$ represents a Khatri-Rao product".

---

> ### Author Rebuttal · Authors · 2025-07-31
>
> We thank the reviewer for their feedback and highlighting the significant improvement in the amount of overparametrization.
>
> We thank the reviewer for their suggestions to improve the presentation, and we will clarify the questions asked in the final version. However, we would like to take this opportunity to clarify some confusion or potential misconceptions. We emphasize that all our claims are rigorously proven in the appendix. The analysis is technically quite involved and does not fit fully within the page limit. In the main body of the paper, we prioritized giving the main intuition by highlighting different challenges in proving the main result and giving a sketch of the main ideas involved, as is the norm in many learning theory papers with technical proofs. The full ArXiv version will include all the proofs in the main body. Now we address some of the specific questions:
>
>  “lines 262-264: "suppose $Y^{(t)}$ and $Z^{(t)}$ were each drawn independently and randomly from $\mathrm{colspan}(A)$, with high probability...": this statement is nontrivial, it would be good to include the reference or proof””
>
> Thanks for the question. The confusion comes from a typo in the sentence.
> Here, the $Z^{(t)}$ should be replaced by $X^{(t)}$. We hope that this clarifies the confusion. Taking that into consideration, the statement is a non-trivial statement about the Khatri-Rao product of random vectors in a subspace (which is actually implied by the much stronger statement in Theorem 4.1). However we emphasize this sentence was merely a hypothetical that was only meant to illustrate the technical challenge. The key point is that such an assumption (about the initialization being random in the $\mathrm{colspan}(A)$ ) cannot be made and our analysis does not rely on this statement in any way. We will correct the typo, and emphasize the above point in the final version.
>
>  “one important part that is missing is to show that with $k=\Omega(r^2)$, the subspace spanned by $A$ will be in the subspace of  $X, Y, Z$ that are randomly generated with high probability.”
>
> There seems to be some misconception here that we would like to clarify. This is not what our proof approach involves; in fact, this should not be expected to be true. For the interesting regime of parameters for r and k, with high probability the spans of the randomly initialized X and Y will have trivial intersection with the column span of A. Here the ALS iteration plays an important role. In fact we prove that after we update X and Y in one iteration of the ALS algorithm, they will be in the subspace spanned by A (this is a statement that follows directly from the expressions of $X^{(1)}$ and $Y^{(1)}$ as the least squares solutions for the problems) but most importantly that $X^{(1)}\odot Y^{(1)}$ spans the subspace spanned by the columns of $A\otimes A$. This is a non-trivial statement.
>
>  “Line 283: "Thus, proving (4) reduces to showing that $X^{(1)}\odot Y^{(1)}$ has rank $r^2$": this statement is nontrivial, it would be good to provide a formal theorem with proof.“
>
> As mentioned already, $X^{(1)}$ and $Y^{(1)}$ ($X$ and $Y$ after the first ALS update) will both be in the subspace spanned by the columns of $A$, this in particular means that $X^{(1)}\odot Y^{(1)}$ will be in the column span of $A\otimes A$. Thus proving that the rank of matrix $X^{(1)}\odot Y^{(1)}$ has rank $r^2$ together with the fact that its columns lie in the subspace spanned by $A \otimes A$ implies that the columns of $X^{(1)}\odot Y^{(1)}$ must span the entire space of $A\otimes A$. We will clarify this in the revised version.
>
> “It’s unclear how to derive the equation below 294, and why is it important.”
>
> Using a standard identity for the Khatri-Rao product (equation (3) in the paper) the matrix $\hat{X}\odot \hat{Y}$ can be written as $(A\otimes A)$ times the matrix below line 299. Using that A is well conditioned (its least singular value being lower bounded), it follows that $A\otimes A$ is also well conditioned using standard multiplicative properties of singular values. Hence arguing about the least singular value of the matrix below line 299 suffices to show our result. As mentioned earlier, we chose to highlight the crucial steps of the proof in the proof overview due to space limitations. The full details of the proof are in the appendix.
>
> To reiterate, we thank you for the questions/ comments. While all of the above technical details and the entire proof are in the appendix, your questions help us by identifying portions of the overview that we can clarify or emphasize better. Please let us know if there are any remaining questions/ clarifications.

---

> > ### Comment · Reviewer_L37m · 2025-08-04
> >
> > I would like to thank authors for the detailed response, which clear a couple of my concerns. This is a great contribution, but it would be good for authors to improve the clarity in the main text of the work. I will raise my score.

---

### Official Review · Reviewer_JGuV · 2025-07-03

**Clarity:** 4
**Significance:** 3
**Originality:** 4
**Rating:** 5
**Confidence:** 3

**Summary:**

This paper analyzed the algorithm of ALS (Alternating Least Squares) with overparameterization for the symmetric low-rank tensor decomposition problem. It was proved that when the overparameterized rank $k \\gtrsim r^2$ ($r$ is the rank of the tensor), randomly initialized ALS converges in two iterations to an approximate optimum. The main ingredient of the proof is a careful analysis of random initialization, in particular, several anti-concentration properties of Gaussian matrices. The paper also included some experiments to discuss whether the $O(r^2)$ dependency is necessary.

**Questions:**

Please refer to the Weaknesses part.

**Ethical Concerns:**

["NO or VERY MINOR ethics concerns only"]

**Final Justification:**

The author has answered my question, and I have no particular problem with this paper right now. I will keep my positive score.

**Limitations:**

yes

**Quality:**

3

**Strengths And Weaknesses:**

The result appears novel and is a welcomed contribution to the study of low (CP-)rank tensor decomposition. Though I have not checked all the details in the proof, I did a few sanity checks and did not find obvious flaws. The experimental results on parallel and sequential ALS are also interesting. I think the paper is overall acceptable.

Speaking of weaknesses, one issue is that the error bound is proven for $X^{(1)} \otimes Y^{(1)} \otimes Z^{(2)}$, rather than $X^{(2)} \otimes Y^{(2)} \otimes Z^{(2)}$. It would be helpful to discuss if anything can be proved for later iterations. Will the error diminish or accumulate?

---

> ### Author Rebuttal · Authors · 2025-07-31
>
> We thank the reviewer for their kind comments about the novelty and significance of the results. The question about the error in later iterations is a nice question, and from a technical standpoint it is related to some of the differences between standard ALS and the parallel version we analyze. Our argument does not imply such a decay in the error, and this seems beyond the scope of our current techniques as far as we can see.

---

### Decision · Program_Chairs · 2025-09-17

**Decision:**

Accept (spotlight)

**Comment:**

The paper analyses the popular alternating least squares (ALS) algorithm for CP tensor decomposition. The paper shows that ALS converges to the global optima if there is sufficient overparameterization: if the tensor has rank r, then ALS with O(r^2) factors converges to a factorization with 0 error. This improves on results of WWL+20 who showed that a variant of gradient descent converges to the global optima with overparameterization O(r^7.5).

The reviewers made some suggestions to improve the writing, and I recommend the authors carefully implement the changes in the final version. There was also a comment about potentially changing the title. On my end, I am okay with the current title for similar reasons to what the authors mentioned, but if the authors prefer to change the title that is fine as well.

Another concern raised was the the paper analyses a variant of ALS. The modification updates all the factors A, B, C in parallel, using their values at the previous iteration, instead of say using the updates value of A to perform the update for B. I consider this modification to be pretty benign in the context of doing modifications to iterative techniques to analyze them.

ALS is by far the most popular algorithm for tensor decomposition, and tensor decomposition has also served as a canonical setting to understand non-convex optimization methods. The paper makes significant contributions to understanding optimization techniques for the problem. There are also some interesting future questions raised here by the reviewers, in particular on the need for overparameterization, and if lower bounds are possible here (the experiments suggest some overparameterization may be necessary). Investigating this further would be very interesting.

--other comments--

The paper is missing citations to some previous work like the following on analyzing ALS for CP decomposition, please correct it:

- Local Convergence of the Alternating Least Squares Algorithm for Canonical Tensor Approximation
- On the Global Convergence of the Alternating Least Squares Method for Rank-One Approximation to Generic Tensors
- Provable Tensor Factorization with Missing Data
- Guaranteed Non-Orthogonal Tensor Decomposition via Alternating Rank-1 Updates